# SPOT: Scalable Policy Optimization with Trees for Markov Decision Processes

**Xuyuan Xiong**[1]     **Pedro Chumpitaz-Flores**[2]     **Kaixun Hua**[*2]     **Cheng Hua**[*1]

[1]Shanghai Jiao Tong University     [2]University of South Florida

## Abstract

Interpretable reinforcement learning policies are essential for high-stakes decision-making, yet optimizing decision tree policies in Markov Decision Processes (MDPs) remains challenging. We propose SPOT, a novel method for computing decision tree policies, which formulates the optimization problem as a mixed-integer linear program (MILP). To enhance efficiency, we employ a reduced-space branch-and-bound approach that decouples the MDP dynamics from tree-structure constraints, enabling efficient parallel search. This significantly improves runtime and scalability compared to previous methods. Our approach ensures that each iteration yields the optimal decision tree. Experimental results on standard benchmarks demonstrate that SPOT achieves substantial speedup and scales to larger MDPs with a significantly higher number of states. The resulting decision tree policies are interpretable and compact, maintaining transparency without compromising performance. These results demonstrate that our approach simultaneously achieves interpretability and scalability, delivering high-quality policies an order of magnitude faster than existing approaches.

## 1 Introduction

In high-stakes or safety-critical domains, it is crucial that reinforcement learning (RL) policies be understandable by humans. Rather than relying on post-hoc explanations of opaque neural policies (e.g., via LIME or SHAP), which can be incomplete or misleading [27], a more direct approach is to learn inherently interpretable policies. Decision tree policies have attracted significant attention as a suitable interpretable model class: they are simple, rule-based decision structures (threshold tests on state features leading to actions) that can represent non-linear behavior while remaining human-comprehensible. A size-limited decision tree (bounded depth or number of leaves) is simulatable by a person (one can manually follow the decision path) and decomposable (each decision node is an understandable rule).

Optimizing a policy constrained to be a decision tree is notoriously difficult [6]. Rudin (2019) [19] identified optimizing sparse logical models such as decision trees as the foremost grand challenge in interpretable machine learning, emphasizing the distinction between inherently interpretable models and post-hoc explanation methods. Unlike differentiable function approximators, decision trees have a discontinuous, non-differentiable structure that precludes straightforward gradient-based training. The space of possible trees grows combinatorially with depth, making brute-force search intractable except for very small cases. In supervised learning, greedy algorithms like CART [8] can find reasonably good trees but are not guaranteed to find the optimal tree and may perform arbitrarily poorly in some cases. In the context of MDPs, an added challenge is that the policy's decisions in one state can influence the distribution of states encountered elsewhere. This means we cannot simply optimize the tree on a static dataset of state-action examples; we must consider the global dynamics of the MDP when evaluating a tree policy. Overall, finding an optimal or near-optimal decision tree

---

[*]Corresponding Authors

39th Conference on Neural Information Processing Systems (NeurIPS 2025).

policy in an MDP is a combinatorial optimization problem on top of the usual RL complexities, and is generally NP-hard. These challenges have motivated a variety of research efforts to learn decision tree policies in a tractable way.

## 1.1 Interpretable Policy Learning in RL

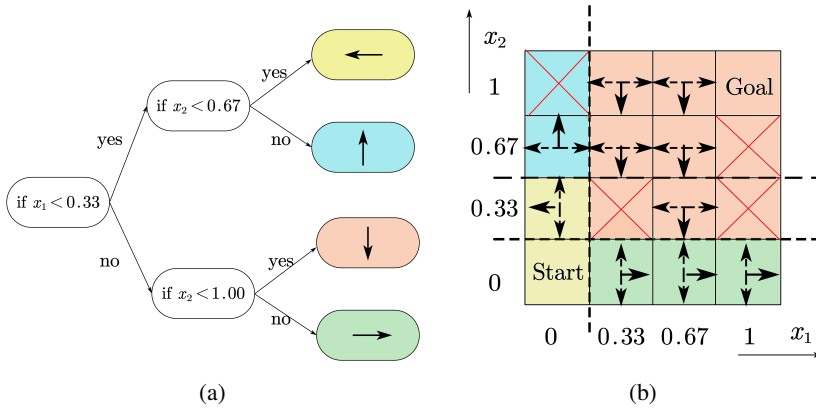

(a)            (b)

Figure 1: Left: An interpretable policy learned by a decision tree of depth $D = 2$. Right: Illustration of the Frozen Lake 4×4 (Stochastic) Markov Decision Process.

A number of prior works have proposed methods to learn interpretable RL policies (decision trees or other rule-based representations) that balance interpretability and performance. These methods can be broadly categorized into: (1) relaxation-based or programmatic methods that directly train a restricted policy, and (2) imitation and distillation methods that extract an interpretable policy from a complex one. We review key representatives of each approach:

One strategy is to soften or relax the tree representation to enable (approximate) gradient-based optimization. For example, Gupta et al. (2015) [12] introduced a policy tree model that is optimized via policy gradient, using a smooth parameterization of the splitting criteria. These approaches embed the non-differentiable tree into a continuous optimization, often achieving decent performance, but they do not guarantee a globally optimal tree. Another innovative idea is to reformulate the RL problem itself to enforce a tree policy: Topin et al. (2021) [22] proposed Iterative Bounding MDPs (IBMDPs), a meta-MDP construction in which any optimal policy corresponds to a decision tree for the original MDP. By solving the IBMDP with standard deep RL algorithms, they indirectly obtain a tree-structured policy. This approach allows using powerful function approximation during training while yielding a discrete tree policy in the end, but the solution quality still depends on the RL training procedure. Finally, beyond decision trees, researchers have explored programmatic policy learning. Verma et al. (2018) [24] introduced Programmatically Interpretable RL (PIRL), using a high-level domain-specific programming language to represent policies. Their method, NDPS (Neurally Directed Program Search), first trains a neural network policy and then performs a guided search in the space of programs to find a policy that mimics the neural policy's decisions.

Another framework for obtaining an interpretable policy is to leverage a teacher, typically a high-performance but complex policy, and train a simpler model to imitate it. Dataset Aggregation (DAgger) [18] is a classic approach where an agent gradually collects states by following its current policy and labels them with actions from an expert policy, iteratively improving the learned policy. VIPER (Verifiable Imitation Policy Extraction) [4] builds on this idea to specifically learn decision tree policies. VIPER augments DAgger by using the teacher's Q-value information to focus the learning on important states. This results in smaller, more accurate decision trees than naive imitation learning. VIPER demonstrated success in distilling deep RL agents (like DQN policies) into compact decision trees that can be formally analyzed. In summary, imitation-based methods can produce high-quality tree policies if the teacher is strong; however, they inherit a fundamental limitation: if the optimal policy in the MDP is too complex to be represented by a small tree, a student forced to imitate that complex optimal (or any complex teacher) will struggle. The student might use its limited capacity to mirror the teacher's decisions in parts of the state space that are actually unnecessary to obtain a high reward. In fact, recent work found that when a limited-depth tree is optimal for

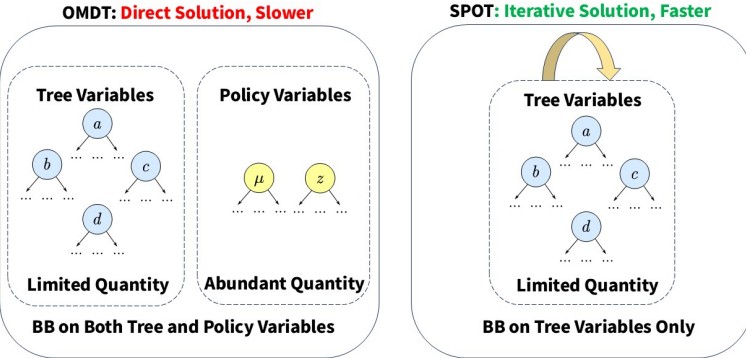

Figure 2: Branching Strategy Comparison between SPOT and OMDT. OMDT uses standard solvers (e.g., Gurobi), requiring branching on all integer variables, thus linearly increasing complexity with the number of states. In contrast, SPOT employs a two-stage reduced-space branch-and-bound approach, branching only on tree variables.

the task, directly optimizing that tree can yield better performance than imitating a complex policy [27]. This highlights an imitation gap: the best interpretable policy may differ from the behavior of a black-box optimal policy, so chasing the latter via imitation can be counterproductive. In light of this, a promising direction is to optimize the decision tree policy directly for the MDP's returns, rather than relying on a teacher.

## 1.2 Optimal Decision Tree Policies

To overcome the performance limitations of approximate methods, researchers have begun to tackle the exact optimization of decision-tree policies in MDPs. Over the past decade, several works have formulated the decision tree induction problem as a mixed-integer program (MIP) or other combinatorial optimization problem to find the globally optimal tree.

Notably, Bertsimas & Dunn (2017) [6] and Verwer & Zhang (2017) [25] introduced MIP models for optimal classification trees, and subsequent extensions incorporated various constraints (fairness [1], robustness to adversarial examples [26], etc.) Due to the NP-hard nature of optimal tree induction, a variety of techniques have been explored to improve solver efficiency: dynamic programming for small trees [10], constraint programming and SAT formulations [14, 16, 20, 23], and specialized branch-and-bound search algorithms that cleverly prune the search space [2, 3]. These works demonstrated that for classification/regression tasks with a fixed dataset, one can often compute an optimal decision tree for modestly sized problems, providing a guarantee of maximal accuracy given the tree size. Vos and Verwer [27] brought this exact optimization approach into the RL setting. OMDT (Optimal MDP Decision Trees) formulates finding a decision-tree policy for a given discrete MDP as a single comprehensive MIP. In essence, their formulation combines the standard linear programming formulation of an MDP's optimal policy with additional integrality constraints that restrict the policy to the structure of a binary decision tree. They link each state's decision to the traversal of some path in the tree and enforce consistency of actions with the tree's predictions. Using a MILP solver, OMDT can directly maximize the expected discounted return of a size-limited decision tree policy. However, the exact approach comes with scalability challenges. Solving a large MILP that encodes the entire MDP and a complex policy structure is computationally intensive. In OMDT's experiments, the approach was feasible for MDPs with on the order of $10^3$–$10^4$ states and for trees of depth up to around 5–7. The authors report that in some environments with larger state spaces (e.g., a tic-tac-toe game MDP), OMDT required hours of runtime to surpass the performance of the approximate VIPER method. This has motivated us to improve the scalability of tree policies.

A key open challenge is how to achieve scalability for decision tree policies. In particular, *can we design algorithms that find the tree policy more efficiently than directly solving the MILP, enabling use on problems with more states?*

In contrast to OMDT's single huge MILP solve, we introduce a decision-tree policy iteration framework that alternates between evaluating the current tree policy and improving it. At each policy improvement step, finding the optimal decision tree (with respect to the current value function) is formulated as an MILP, but we solve it using a reduced-space branch-and-bound procedure, which leverages the problem's structure by decoupling the MDP constraints. By distributing our approach across multiple processors, we exploit parallelism to further speed up the search for the optimal tree policy in each iteration. Empirically, in our experiments, this approach outperforms OMDT in both runtime and scalability, finding optimal or near-optimal tree policies in problems where the baseline MILP approach fails or takes prohibitively long.

Our work addresses gaps in interpretable reinforcement learning by developing a scalable and efficient algorithm for optimizing interpretable decision-tree policies in Markov Decision Processes (MDPs). The key contributions of this work are summarized as follows:

- The proposed method produces compact, interpretable policies suitable for deployment in sensitive, safety-critical domains.
- We propose a novel decomposition strategy inspired by policy iteration. Instead of directly solving a monolithic optimization, we iteratively decompose the problem into smaller, more tractable subproblems. We provide theoretical guarantees that each of these subproblems can be solved to global optimality, ensuring overall high-quality solutions.
- By leveraging the decomposition strategy, the optimization problems arising in each iteration become independently solvable and highly parallelizable. This significantly enhances computational efficiency and scalability.
- Extensive numerical experiments on standard benchmarks demonstrate that our approach yields solutions dramatically faster than state-of-the-art methods. We observe significant runtime improvements and demonstrate the capability to efficiently handle larger problem instances.

## 2   Preliminaries

**Markov Decision Processes.**    A Markov Decision Process (MDP) provides a mathematical framework for modeling sequential decision-making problems under uncertainty. Formally, an MDP is defined as a tuple $(\mathcal{S}, \mathcal{A}, P, R, \gamma)$, where $\mathcal{S}$ is a finite set of states, $\mathcal{A}$ is a finite set of actions, $P_{ii'k}$ represents the transition probability from state $i$ to state $i'$ given action $k$, $R_{ii'k}$ denotes the immediate reward received for transitioning from state $i$ to $i'$ under action $k$, and $\gamma \in [0, 1)$ is the discount factor. The goal is to find a policy $\pi$ that maximizes the expected cumulative discounted reward $\mathbb{E}_\pi \left[ \sum_{t=0}^\infty \gamma^t R_{i_t i_{t+1} k_t} \right]$.[2]

**Solving MDPs via Linear Programming and its Dual Formulation.**    For a finite, discounted MDP, an optimal policy can be computed by formulating and solving a linear programming (LP) problem. Let $V_i$ denote the optimal value function at state $i$, let $p_0(i)$ represent the probability of starting in state $i$, and let $P_{ii'k}$ and $R_{ii'k}$ denote, respectively, the transition probability and immediate reward when action $k$ is executed in state $i$ transitioning to state $i'$. The primal LP formulation directly encodes the Bellman optimality conditions in terms of state values:

$$\min_V \quad \sum_{i \in \mathcal{S}} p_0(i)\, V_i$$

$$\text{s.t.} \quad V_i \; - \; \gamma \sum_{i' \in \mathcal{S}} P_{ii'k}\, V_{i'} \; \geq \; \sum_{i' \in \mathcal{S}} P_{ii'k}\, R_{ii'k}, \quad \forall\, i \in \mathcal{S},\, k \in \mathcal{A}.$$

At optimality, these constraints hold with equality for at least one action per state. Intuitively, the primal LP minimizes each state's value while satisfying the constraints derived from Bellman's equation, resulting in the optimal value function. Subsequently, an optimal policy can be derived by selecting the action(s) in each state that yield equality in the corresponding constraints.

Taking the dual of the primal LP yields an alternative formulation where the variables explicitly represent the frequency with which each state-action pair is utilized. Define $\varphi_{ik} \geq 0$ as the *discounted occupancy measure*, representing the expected discounted number of times the agent visits state $i$

---

[2]We use $i$ to represent state and $k$ to represent action, differing from the traditional notation of $s$ and $a$. This choice is made to distinguish from variables commonly used in decision-tree contexts.

and selects action $k$. The dual LP maximizes the total expected discounted reward subject to linear *flow-balance* constraints:

$$\max_{\varphi} \quad \sum_{i \in \mathcal{S}} \sum_{k \in \mathcal{A}} \varphi_{ik} \left( \sum_{i' \in \mathcal{S}} P_{ii'k} R_{ii'k} \right)$$

$$\text{s.t.} \quad \sum_{k \in \mathcal{A}} \varphi_{ik} = p_0(i) + \gamma \sum_{i' \in \mathcal{S}} \sum_{k \in \mathcal{A}} P_{i'ik} \varphi_{i'k}, \quad \forall i \in \mathcal{S}. \tag{1}$$

In this formulation, the term on the left-hand side of each constraint represents the *flow out* of state $i$, while the right-hand side represents the *flow into* state $i$, consisting of contributions from the initial state distribution and transitions from predecessor states. Any feasible solution $\mu$ thus corresponds to a valid policy, and the dual objective directly quantifies the expected return of this policy.

# 3 Scalable Policy Optimization with Trees

The primary challenge in solving the full dual optimization problem for tree-structured MDP policies arises from the coupling introduced by the MDP transition dynamics. Specifically, the system of equations defined by Eq. (1) creates a fully interconnected set of constraints, wherein each state's decision variables depend on transition probabilities that link them to all other states. This interconnectedness implies that the optimization problem is not decomposable; it cannot be divided into smaller, independently solvable subproblems because the transition terms inherently couple all state-action variables together. Consequently, directly solving the complete dual formulation becomes computationally impractical.

To address this issue, we draw inspiration from the classical value iteration algorithm [5, 13]. Rather than attempting to solve the entire coupled system simultaneously, we adopt an iterative strategy. We start with an initial estimate of the value function, denoted by $V^{\text{old}}$, and perform a single-step Bellman update for each state based on this fixed value. For each state $i$, the single-step Bellman update can be formulated as:

$$\min_{V_i} \quad V_i$$

$$\text{s.t.} \quad V_i - \gamma \sum_{i' \in \mathcal{S}} P_{ii'k} V_{i'}^{\text{old}} \geq \sum_{i' \in \mathcal{S}} P_{ii'k} R_{ii'k}, \quad \forall k \in \mathcal{A}. \tag{2}$$

It can be found that iteratively solving Eq. (2) corresponds exactly to performing value iteration. The critical advantage of adopting this iterative approach in our formulation is its inherent decomposability. Specifically, in the Bellman backup LP presented above, the constraints are separated by state. Each state $i$ is associated with its own independent set of inequalities, which eliminates direct coupling and interdependencies between states.

Analyzing the dual form of this single-step LP provides insights into the underlying policy updates. Specifically, the dual formulation of Eq. (2) for state $i$ can be expressed as

$$\max_{\mu} \quad \sum_{k \in \mathcal{A}} \mu_{ik} \sum_{i' \in \mathcal{S}} P_{ii'k} (R_{ii'k} + \gamma V_{i'}^{\text{old}})$$

$$\text{s.t.} \quad \sum_{k \in \mathcal{A}} \mu_{ik} = 1,$$

$$\mu_{ik} \geq 0, \quad \forall k \in \mathcal{A},$$

where $\mu_{ik}$ is the dual variable which also considered as the *policy*. When extending this policy update formulation jointly across all states $\mathcal{S}$, we arrive at the following aggregated optimization problem

$$\max_{\mu} \quad \sum_{i \in \mathcal{S}} \Phi_i^{\pi^{\text{old}}} \sum_{k \in \mathcal{A}} \mu_{ik} \sum_{i' \in \mathcal{S}} P_{ii'k} (R_{ii'k} + \gamma V_{i'}^{\text{old}}) \tag{3a}$$

$$\text{s.t.} \quad \sum_{k \in \mathcal{A}} \mu_{ik} = 1, \quad \forall i \in \mathcal{S}, \tag{3b}$$

$$\mu_{ik} \geq 0, \quad \forall i \in \mathcal{S}, k \in \mathcal{A}. \tag{3c}$$

Here, the coefficient $\Phi_i^{\pi^{\text{old}}}$ serves as a weight, indicating the relative importance given to each state $i$ during the overall policy update. This conceptualization is further clarified in Proposition 1.

Now, we introduce the decision tree policy constraints. Consider that each state $i$ has an associated feature vector $x_i \in \mathbb{R}^F$, with each feature scaled to the unit interval $[0, 1]$. The decision tree consists of nodes indexed by $t = 1, \ldots, T$, where $T = 2^{D+1} - 1$ denotes the total number of nodes for a tree of depth $D$. Following the notation from [15], we let $p(t) = \lfloor t/2 \rfloor$ denote the parent of node $t$, and define the sets $A_L(t)$ and $A_R(t)$ to include the ancestors of node $t$ where the left or right branch, respectively, is taken along the path from the root to node $t$. The tree nodes are partitioned into decision nodes $\mathcal{T}_B = \{1, \ldots, \lfloor T/2 \rfloor\}$ and leaf nodes $\mathcal{T}_L = \{\lfloor T/2 \rfloor + 1, \ldots, T\}$. The resulting constraints for the decision-tree-structured policy optimization problem are as follows:

**Tree Constraints**

$$\sum_{k \in \mathcal{A}} c_{kt} = 1, \qquad \forall t \in \mathcal{T}_L \tag{3d}$$

$$\sum_{t \in \mathcal{T}_L} z_{it} = 1, \qquad \forall i \in \mathcal{S} \tag{3e}$$

$$a_m^T(x_i + \epsilon - \epsilon_{\min}) + \epsilon_{\min} \le b_m + (1 + \epsilon_{\max})(1 - z_{it}), \qquad \forall t \in \mathcal{T}_L, m \in A_L(t), i \in \mathcal{S} \tag{3f}$$

$$a_m^T x_i \ge b_m - (1 - z_{it}), \qquad \forall t \in \mathcal{T}_L, m \in A_R(t), i \in \mathcal{S} \tag{3g}$$

$$\sum_{j=1}^{F} a_{jt} = d_t, \qquad \forall t \in \mathcal{T}_B \tag{3h}$$

$$0 \le b_t \le d_t, \qquad \forall t \in \mathcal{T}_B \tag{3i}$$

$$d_t \le d_{p(t)}, \qquad \forall t \in \mathcal{T}_B \tag{3j}$$

**Binary Constraints**

$$a_{jt}, d_t \in \{0, 1\}, \qquad \forall t \in \mathcal{T}_B \tag{3k}$$

$$z_{it} \in \{0, 1\}, \qquad \forall t \in \mathcal{T}_L \tag{3l}$$

$$c_{kt} \in \{0, 1\}, \qquad \forall k \in \mathcal{A}, t \in \mathcal{T}_L \tag{3m}$$

**Policy-Tree Coupling Constraint**

$$z_{it} + c_{kt} - 1 \le \mu_{ik}, \qquad \forall i \in \mathcal{S}, k \in \mathcal{A}, t \in \mathcal{T}_L \tag{3n}$$

In alignment with the constraint-design framework from [15], the tree structure is encoded through four sets of variables: $a, b, c, d$. At each decision node $t \in \mathcal{T}_B$, the binary variable $d_t$ indicates whether a node splits. When a split occurs, it is characterized by a binary feature selection vector $a_t = [a_{1t}, \ldots, a_{Ft}]^\top \in \{0, 1\}^F$ and a threshold $b_t \in [0, 1]$. At the leaves, $c_{kt}$ specifies the action $k$ assigned to leaf $t$, and $z_{it}$ records if state $i$ terminates at leaf $t$. Constants $\epsilon, \epsilon_{\max}, \epsilon_{\min}$ stabilize the mixed-integer big-M calculations as described in [7] (See also Appendix A).

These constraints collectively preserve the logical and structural coherence of the decision tree. Specifically, Constraint (3d) ensures that each leaf node is associated with exactly one action label. Constraint (3e) guarantees that each state is assigned uniquely to one leaf node. Constraints (3f) and (3g) implement the appropriate branching behavior, directing states correctly based on their feature values. Constraints (3h) and (3i) ensure proper handling of non-splitting nodes: when no split occurs at a node (i.e., $d_t = 0$), all states reaching this node follow the right branch, ultimately directing them consistently towards the right-most leaf. Constraint (3j) maintains hierarchical consistency, enforcing that splits at parent nodes precede splits at child nodes, thereby ensuring logical downward propagation of tree branches. Finally, Constraint (3n) explicitly restricts the feasible policy space to policies that can be represented by decision trees.

Next, we explicitly derive $\Phi^{\pi^{\text{old}}}$. This quantity is the *discounted occupancy measure*, representing the expected cumulative number of discounted visits to each state under the current policy $\pi^{\text{old}}$. Formally, it is defined as:

$$\Phi_i^{\pi^{\text{old}}} = \sum_{t=0}^{\infty} \gamma^t \mathbb{P}(s_t = i | \pi^{\text{old}})$$

**Algorithm 1:** Scalable Policy Optimization with Trees (SPOT)

---

**Input:** Number of iterations $n$, a sequence of thresholds $\{\phi_l\}_{l=1}^n$

1   Initialize random policy $\pi_0$ and compute its value function $V^0$;

2   Set $V^{\text{old}} \leftarrow V^0$;

3   **for** $l = 1$ *to* $n$ **do**

4      **Fix:** Select tree parameters as tunable with probability $\phi_l$; fix others;

5      **Solve:** With $V^{\text{old}}$, solve Problem (3) via Algorithm 2 to obtain policy $\pi_l$;

6      **Evaluate:** Compute $V_l$ and state distribution $\Phi^{\pi_l}$ for $\pi_l$;

7      **Update:** $V^{\text{old}} \leftarrow V_l$, $\Phi^{\pi^{\text{old}}} \leftarrow \Phi^{\pi_l}$;

8   **end**

**Output:** The policy in $\{\pi_1, \ldots, \pi_n\}$ with highest expected return

---

where $\mathbb{P}(s_t = i \mid \pi^{\text{old}})$ denotes the probability of being in state $i$ at time step $t$ when following policy $\pi^{\text{old}}$. The complete Algorithm is presented in Algorithm 1.

**Proposition 1.** *Each iteration of SPOT corresponds to performing a one-step policy gradient ascent update starting from $\pi^{old}$. In particular, the update maximizes the same first-order objective that the policy gradient theorem uses, resulting in a new policy that is a greedy improvement on $\pi^{old}$.*

SPOT differs from traditional policy gradient methods in a key way: instead of taking a small update step in the gradient direction, we fully re-optimize the policy $\mu$ to maximize the surrogate linearized objective around $\pi^{\text{old}}$. This can be interpreted as performing exact policy improvement rather than incremental improvement. Therefore, one iteration of SPOT corresponds to a one-step greedy policy improvement that leverages the structure of the policy gradient objective while optimizing it completely under the current value estimate of $\pi^{\text{old}}$.

**Proposition 2.** *If the class of decision-tree policies is expressive to represent an optimal MDP policy, and the weighting vector $\Phi^{old}$ is strictly positive, then SPOT is guaranteed to converge to the optimal solution.*

## 4   Reduced-Space Branch-and-Bound Framework

The optimization problem described by (3) can be viewed as a two-stage stochastic programming (TSSP) problem. Although traditional off-the-shelf solvers are generally effective, directly solving these large-scale problems is often computationally intensive or even intractable in practice.

To address this challenge, we propose a reduced-space branch-and-bound (RSBB) framework specifically designed for solving mixed-integer TSSP problems, enhanced with carefully constructed lower and upper bounds [9]. This RSBB procedure initiates with the full feasible region $M_0$ and iteratively refines the optimality gap by bisecting $M_0$ into progressively smaller subsets, discarding any subregions that cannot contain the optimal policy. At each iteration, a subregion $M$ is assessed by computing a lower bound $\beta(M)$ and an upper bound $\alpha(M)$. If the gap between these bounds satisfies $\alpha(M) - \beta(M) \leq \delta$, where $\delta$ represents an extremely small threshold, the algorithm terminates. Otherwise, the subregion $M$ is further partitioned according to a defined branching rule, and the process continues iteratively.

A significant advantage of this RSBB approach is its proven convergence, which occurs despite branching exclusively on tree-structure descriptive variables, such as the split-indicator and threshold vectors (i.e., $a, b, c, d$ in constraints of Problem (3)). This convergence property remains valid [15] even in the presence of constraints involving both continuous and binary second-stage variables.

**Theorem 1.** *RSBB converges to the global optimum, formally expressed as*

$$\lim_{i \to \infty} \alpha_i = \lim_{i \to \infty} \beta_i = f. \tag{4}$$

Here, $f$ indicates the complete MILP formula of Problem (3). The finite nature of the feasible region for optimal policies arises naturally from the finite number of distinct tree structures that can result from the discrete selection of thresholds $b$ at each decision node.

### 4.1 Two Stage Stochastic Program

Note that variables $a, b, c, d$ describe the structure of the decision tree and are the same for all states, while policy-related variables $z$, and $\mu$ are state-specific and describe the allocation and reward of a specific state. Therefore, Problem (3) with fixed $V^{\text{old}}$ can be reformulated as the following formula:

$$f(M_0) = \max_{m \in M_0} \sum_i Q_i(m), \tag{5}$$

where we denote $m = (a, b, c, d)$ as all first-stage variables and $Q_i(\cdot)$ represents the optimal value of the second-stage problem:

$$Q_i(m) = \max_{z_i, \mu_i} \Phi_i^{\pi^{\text{old}}} \sum_k \mu_{ik} \sum_{i'} P_{ii'k} \left( R_{ii'k} + \gamma V_{i'}^{\text{old}} \right) \tag{6a}$$

$$\text{s.t. Constraints (3a), (3d)} \sim \text{(3n)}. \tag{6b}$$

The closed set $M_0 := [m^l, m^u]$ denotes the region of the first stage variables. $(\cdot)^l, (\cdot)^u$ represent the lower and upper bounds of each variable. In each branch-and-bound (BB) node with a specific partition $M \subseteq M_0$, we solve the problem $f(M) = \max_{m \in M} \sum_{i \in \mathcal{S}} Q_i(m)$.

Leveraging this problem reformulation and its decomposable structure, we can derive effective lower and upper bounding strategies within the RSBB framework to solve Problem (5). The lower bounds can be directly constructed using existing techniques from the optimal decision tree literature [15], including scenario grouping, relaxed mixed-integer linear programming (MILP) bounds, and simple primal bound searches. Unlike traditional formulations for optimal decision tree policy learning, the decomposable structure of our approach enables straightforward parallelization, which significantly accelerates convergence.

However, to fully leverage the unique structure and decomposability of Problem (5), we introduce a tailored closed-form decomposable upper bounding strategy specifically designed for this maximization problem. Additionally, this approach allows us to implement effective bound-tightening techniques that accelerate convergence within the RSBB framework by pre-determining the optimal actions for certain states during the branch-and-bound process. The overall structure and procedure of the RSBB algorithm are summarized in Algorithm 2 (see Supplementary Material).

---

**Algorithm 2:** Reduced-Space Branch-and-Bound for Optimal Tree Policy

**Input:** $M_0$, non-zero tolerance $\epsilon$

1 Set iteration index $i = 0$, $\mathbb{M} \leftarrow \{M_0\}$ ;
2 Initial upper and lower bounds $\alpha_i = \alpha(M_0)$, $\beta_i = \beta(M_0)$;
3 **repeat**
4    **Node Selection**
5       Select a set $M \in \mathbb{M}$ satisfying $\beta(M) = \beta_i$;
6       $\mathbb{M} \leftarrow \mathbb{M} \setminus \{M\}$;
7       $i \leftarrow i + 1$;
8    **Branching**
9       Partition $M$ into subsets $M_1$ and $M_2$ according to the branch strategy in Section E;
10       Add each subset to $\mathbb{M}$ to create separated descendent nodes;
11    **Bounding**
12       Compute $\alpha(M_1), \beta(M_1), \alpha(M_2), \beta(M_2)$;
13       If $\beta_s(M_j), j \in \{1, 2\}$ is infeasible for some $s \in \mathcal{S}$, $\mathbb{M} \leftarrow \mathbb{M} \setminus \{M_j\}$;
14       $\beta_i \leftarrow \max\{\beta(M') \mid M' \in \mathbb{M}\}$;
15       $\alpha_i \leftarrow \max\{\alpha_{i-1}, \alpha(M_1), \alpha(M_2)\}$;
16       Remove all $M'$ from $\mathbb{M}$ if $\alpha(M') \leq \beta_i$;
17       If $|\beta_i - \alpha_i| \leq \delta$, STOP;
18 **until** $\mathbb{M} = \emptyset$;

---

### 4.2 Closed-Form Upper Bounding Strategy

At each BB node, the optimization problem includes an implicit constraint known as the non-anticipativity constraint within the stochastic programming literature. This constraint requires that

all states share the same first-stage variables (i.e., the decision tree structure). By relaxing this non-anticipativity constraint, we derive an upper bounding problem defined as

$$\alpha(M) := \max_{m_i \in M} \sum_{i \in \mathcal{S}} Q_i(m_i). \tag{7}$$

This relaxed problem naturally decomposes into $|\mathcal{S}|$ independent subproblems: $\alpha_i(M) := \max_{m \in M} Q_i(m)$ with $\alpha(M) := \sum_{i \in \mathcal{S}} \alpha_i(M)$. The optimal value $\alpha_i(M)$ for each state $i$ can be efficiently computed by enumerating all possible leaf nodes that state $i$ could reach, without explicitly solving any optimization problems. The computational complexity of this enumeration approach is $O\left(|\mathcal{T}_B| + |\mathcal{T}_L|\right)$. Since decision trees typically maintain a small depth for interpretability, this calculation can substantially outperform traditional optimization methods. Given a bound $M = \left[m^l, m^u\right]$, the enumeration process exhaustively identifies all feasible leaf nodes for state $i$, thereby determining the globally optimal value of $\alpha_i(M)$.

**State Action Pre-determination** An important benefit of the enumeration process utilized in the closed-form upper bound calculation is the reduction of feasible leaf nodes for each state $i$ from the full set $\mathcal{T}_L$ to a smaller set $\mathcal{T}_{z_i}$. Here, $\mathcal{T}_{z_i}$ denotes the set of leaf nodes reachable by state $i$ given the current subregion $M$. By comparing the action selection indicators $c_{kt}$ with their associated range (i.e., $c_{kt}^l, c_{kt}^u$) across each leaf node $t \in \mathcal{T}_{z_i}$, it becomes possible to pre-determine the optimal action and reward for certain states even before solving their corresponding subproblems explicitly. Specifically, the reward and optimal action for state $i$ under the current branch-and-bound (BB) node can be evaluated by, for any $k \in \mathcal{A}, i \in \mathcal{S}$,

$$R_i = \begin{cases} Q_{ik} & \text{if } \bigwedge_{t \in T_{z_i}} c_{kt}^l = c_{kt}^u = 1 \\ \perp & \text{otherwise}, \end{cases} \tag{8}$$

where $\perp$ means the upper bound is not determined.

Once the optimal action and corresponding value for a specific state $i$ have been determined at a node within the BB algorithm, this state can be excluded from subsequent upper-bound calculations. This effectively reduces the computational load by decreasing the number of states to consider. Importantly, these predetermined state-action pairs remain valid throughout all subsequent descendant nodes in the BB search tree. This proactive state-action determination significantly mitigates computational complexity, especially at deeper levels of the search tree. By avoiding redundant optimization subproblems for previously determined states, the overall efficiency and scalability of the algorithm are markedly enhanced.

## 5 Experiments

We evaluate our algorithm by comparing it to OMDT [27] on large-scale MDPs.

**MDP Preparation** We prepare a set of 9 MDP datasets to evaluate our methods: `sys_ad_1`, `sys_ad_2`, `tic_vs_ran`, `tiger_vs_ant`, `csma_2_2`, `csma_2_4`, `firewire`, `wlan0`, and `wlan1`. The first four MDPs are available at `https://github.com/tudelft-cda-lab/OMDT`, while the remaining five can be found at `https://github.com/prismmodelchecker/prism-benchmarks/`. The properties of MDPs are shown in Table 1.

**Comparison** We evaluate two variants of our method: (1) direct application of SPOT (Algorithm 1), and (2) SPOT with a warm start, referred to as SPOT+WS. In SPOT+WS, an initial solution generated by OMDT under a 5-minute time constraint serves as a warm start, subsequently refined by SPOT. All experiments were performed with a uniform 60-minute computational budget. Table 1 reports the performance results for both variants, including the warm-start baselines.

Table 1 summarizes the normalized returns for 9 benchmark MDPs. SPOT+WS consistently outperforms both vanilla SPOT and OMDT, achieving notable improvements (e.g., 1201.7% on `firewire` and 505.95% on `csma_2_2`). Furthermore, our analysis reveals that the warm start strategy is particularly effective for SPOT on larger MDPs, such as `firewire`, `csma_2_4`, and `wlan0`. Even when OMDT already yields strong results (e.g., `wlan0`), SPOT maintains comparable performance.

Table 1: Comparison of OMDT and SPOT on Large MDPs with Tree Depth $D = 3$. Values represent normalized returns. The gain is calculated as SPOT and SPOT+WS gains relative to the absolute value of OMDT (60min). The threshold is set to $\phi_l = 0.5$

| MDP | $|\mathcal{S}|$ | $|\mathcal{A}|$ | F | OMDT (5 min) | OMDT (60 min) | SPOT (60 min) | SPOT+WS (60 min) | SPOT Gain (%) | SPOT+WS Gain (%) |
|---|---|---|---|---|---|---|---|---|---|
| sys_ad_1 | 256 | 9 | 8 | 0.90884 | 0.94064 | 0.94260 | 0.93861 | **+0.2084** | -0.2158 |
| sys_ad_2 | 256 | 9 | 8 | 0.60547 | 0.77446 | 0.80902 | 0.80902 | **+4.4625** | **+4.4625** |
| tic_vs_ran | 2424 | 9 | 27 | -1.0170 | -1.0143 | -1.0081 | -1.0158 | **+0.6113** | -0.1479 |
| tiger_vs_ant | 625 | 5 | 4 | 0.47150 | 0.81333 | 0.84102 | 0.65309 | **+3.4045** | -19.7016 |
| csma_2_2 | 1038 | 8 | 11 | 1.2025e-4 | 4.7332e-4 | 2.3898e-4 | 2.8681e-3 | -49.5165 | **+505.95** |
| csma_2_4 | 7958 | 8 | 11 | -1.1582e-2 | -1.1582e-2 | -1.1699e-2 | 1.1466e-4 | -1.0103 | **+100.99** |
| firewire | 4093 | 13 | 10 | -9.9350e-3 | 4.8978e-2 | -1.0035e-2 | 0.63757 | -120.4926 | **+1201.7** |
| wlan0 | 2954 | 6 | 13 | 0.93399 | 1.00000 | 0.99558 | 1.00000 | -0.4420 | **0.0000** |
| wlan1 | 6825 | 6 | 13 | -0.76336 | 0.98730 | 0.99694 | 0.99694 | **+0.9764** | **+0.9764** |

## 6 Conclusion

In this paper, we presented a novel framework, SPOT, designed for computing decision tree policies in Markov Decision Processes (MDPs). Our approach addresses the critical computational bottlenecks encountered when optimizing tree-structured policies through policy iteration and a reduced-space branch-and-bound algorithm. SPOT provides solid improvements on challenging MDPs while maintaining competitive performance on simpler instances

Despite these results, our method has some limitations. Its performance still relies on the efficiency of the underlying optimization solver, which may restrict scalability for extremely large or complex MDPs. Future research could explore solver-independent acceleration methods or approximate formulations to extend applicability to very large-scale decision-making problems.

## Acknowledgments and Disclosure of Funding

The authors are grateful to Wenhui Zhao for his valuable feedback. Cheng Hua is partly supported by the National Natural Science Foundation of China (72301172, 72394370:72394375, 72495130:72495132), Shanghai Education Commission Chenguang Program (22CGA12), and Shanghai Jiao Tong University Office of Liberal Arts (ZHWK2502).

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

## A  Notation

We now clarify the symbols we used in writing:

Table 2: Summary of Notation

| Category | Description | Example Variables |
|---|---|---|
| Decision Variables | Optimization variables to be solved | $a_{jt} \in \{0, 1\}, b_t \in [0, 1], \mu_{ik} \in [0, 1]$ |
| MDP Parameters | Fixed transition probabilities and rewards | $P_{ii'k}, R_{ii'k}, \gamma$ |
| Algorithm Constants[3] | Fixed hyperparameters and computed values | $\epsilon, \epsilon_{max}, \epsilon_{min}, \Phi_i^{\pi^{old}}$ |
| State/Action Indices | Index sets for states and actions | $i, i' \in \mathcal{S}, k \in \mathcal{A}$ |
| Time Indices | Temporal or iteration indices | $t$ |

## B  Case study: `tiger_vs_ant`

The `tiger_vs_ant` environment is a 5×5 grid world in which a tiger pursues an antelope. The state is the tuple antelope_x, antelope_y, tiger_x, tiger_y, with features normalized to [0,1]. The tiger can move up, right, down, left, or wait. The antelope moves randomly among valid cells, never leaving the grid or stepping onto the tiger's current cell. The agent receives a reward of 1 when the tiger catches the antelope (same cell) and 0 otherwise, and the episode ends upon capture, and rewards are discounted with a factor $\gamma \in (0, 1)$. The task is challenging because the agent must anticipate stochastic antelope motion and use positioning to restrict escape routes

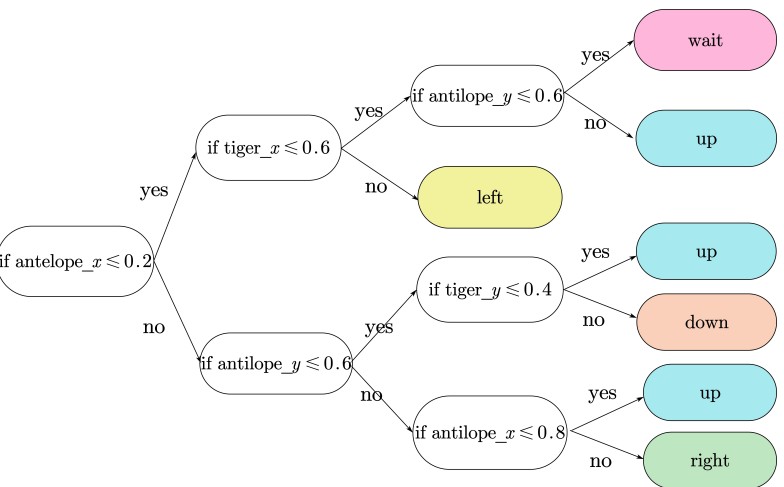

Figure 3: Learned interpretable decision tree policy with $D = 3$ in environment `tiger_vs_ant`

We will demonstrate the interpretability of the above learned policy through the following aspects:

**1. Domain-Meaningful Behavior (which shows the strategic coherence)**

From the learned decision tree policy, we summarize the following strategies:

**Cornering Strategy**: When antelope is trapped near the left wall ($x \le 0.2$), the tiger adapts:

---

[3]$\epsilon, \epsilon_{max}, \epsilon_{min}$ are introduced due to the Big-M reformulation on tree direction constraints. Here $\epsilon$ is a vector of length $F$ (i.e., the number of state features), where for each state feature $j$, we define $\epsilon_j$ as follow [7]: $\epsilon_j = \min\{x_{(i+1),j} - x_{i,j} | x_{(i+1),j} \ne x_{i,j}, i \in [|\mathcal{S}| - 1]\}$, where $x_{i,j}$ denotes the $i$th largest value in the $j$th feature. $\epsilon_{min} = \min_j\{\epsilon_j\}$ and $\epsilon_{max} = \max_j\{\epsilon_j\}$.

- If tiger is well-positioned ($x \le 0.6$), it waits patiently or chases vertically
- If tiger is far away ($x > 0.6$), it prioritizes closing the horizontal gap

**Pursuit Strategy:** In open areas, the tiger uses intelligent vertical positioning:

- Moves upward when positioned below the prey (tiger_y $\le 0.4$)
- Moves downward when positioned above the prey (tiger_y $> 0.4$)

**Escape Prevention**: Near top boundary (antelope_y $> 0.8$), the policy focuses on blocking escape routes rather than direct pursuit

**2. Addresses Complexity Concern** (This pattern correlates to the concept of Simulatability in [11])

- Only 8 decision rules cover the entire 625-state space ($5^4$ states)
- Each path through the tree corresponds to a clear strategic situation
- The shallow depth ($D = 3$) ensures human comprehension

**3. Uses Interpretable Features** (This pattern correlates to the concept of Decomposability in [11])

- Spatial boundaries: Recognizes walls ($x \le 0.2$ for left edge, $y > 0.8$ for top)
- Relative positioning: Implicitly reasons about tiger-antelope spatial relationships
- No complex engineered features: Uses raw normalized coordinates that directly map to grid positions.

## C   Additional Experiment Details

### C.1   Experiment Details

**Details of SPOT Implementation.** During SPOT training, we adopt a time-constrained setup with a total runtime budget of one hour (60 minutes) and a maximum of five minutes allocated per iteration. At each step, we retain the best solution found within the allotted time window. We fix the threshold $\phi_l$ at 0.5 and, for simplicity, set all coefficients $Phi_i^{\pi_{\text{old}}}$ to 1. To enhance exploration, we employ an epsilon-decay strategy: when computing $V^{\text{old}}$, the evaluation is based on a greedy policy with an exploration probability $\epsilon = 1/l$, where $l$ denotes the current iteration index. With probability $\epsilon$, the policy randomly selects alternative actions rather than the greedy one. This mechanism encourages the policy to explore beyond the most immediate choices. In each iteration, we use Algorithm 2 to compute the solution.

### C.2   Experiments with Tree Depth $D = 4$

**Setup.** The MDP setup and method comparison follow those in Section 5. The SPOT implementation details are identical to Section C.1, except that the tree-depth parameter is now set to $D = 4$.

**Results.** Table 3 reports the normalized return across all benchmarks. We find that even the vanilla SPOT method surpasses OMDT in seven out of nine instances—for example, it increases the return on `sys_ad_2` from 0.75403 to 0.83337 (+10.76%) and reduces the loss on `tic_vs_ran` from –1.00610 to –1.00089. Incorporating warm-start initialization further amplifies these improvements: on `csma_2_2`, SPOT+WS boosts the return from 0.00024 to 0.01485 (+6101.14%), and on `firewire`, from –0.009935 to 0.63699 (+6514.90%). Even in the moderate case of `csma_2_4`, warm starts boost performance by +100.97% (–0.01158 → 0.01147). These results confirm that SPOT's depth tuning yields steady improvements and that warm-start strategies can unlock substantial additional performance gains in more challenging MDPs.

### C.3   Experiments on Running Time

To demonstrate SPOT's efficiency in solving each iteration problem, we conduct experiments comparing the runtime of different solution methods on the same optimization task.

Table 3: Comparison of OMDT and Our Methods on Large MDPs with Tree Depth $D = 4$. Values are normalized returns.

| MDP | $|\mathcal{S}|$ | $|\mathcal{A}|$ | $F$ | OMDT (5 min) | OMDT (60 min) | SPOT (60 min) | SPOT+WS (60 min) | SPOT Gain (%) | SPOT+WS Gain (%) |
|---|---|---|---|---|---|---|---|---|---|
| sys_ad_1 | 256 | 9 | 8 | 0.90823 | 0.94403 | 0.95416 | 0.95027 | **+1.073** | +0.6610 |
| sys_ad_2 | 256 | 9 | 8 | 0.74003 | 0.75403 | 0.83337 | 0.83519 | +10.52 | **+10.76** |
| tic_vs_ran | 2424 | 9 | 27 | -1.01700 | -1.00610 | -1.00089 | 0.38958 | +0.5178 | **+138.7** |
| tiger_vs_ant | 625 | 5 | 4 | 0.53266 | 0.80464 | 0.92347 | 0.81389 | **+14.77** | +1.150 |
| csma_2_2 | 1038 | 8 | 11 | 2.3902e-4 | 2.3944e-4 | -1.2252e-2 | 1.4848e-2 | -5216.9 | **+6101.14** |
| csma_2_4 | 7958 | 8 | 11 | -1.1582e-2 | -1.1582e-2 | 1.1466e-4 | -1.1582e-2 | **+100.99** | 0.0000 |
| firewire | 4093 | 13 | 10 | -9.9350e-3 | -9.9350e-3 | 0.63699 | 0.63757 | +6511.6 | **+6517.4** |
| wlan0 | 2954 | 6 | 13 | 0.42805 | 1.00000 | 0.97792 | 0.97792 | -2.208 | -2.208 |
| wlan1 | 6825 | 6 | 13 | -0.76973 | 0.37665 | 0.99694 | 0.99694 | **+164.68** | **+164.68** |

**Setup.** For each MDP, we initialize the policy by selecting a fixed action across all states. Based on this fixed policy, we compute the corresponding value function $V^{\text{old}}$ and set $\Phi^{\pi^{\text{old}}}$ to the softmax of the discounted state-action occupancy measure. We then perform a single iteration of optimization to find a tree with depth $D = 3$ under three different approaches: (i) solving directly with Gurobi, (ii) using RSBB (reduced space branch-and-bound) in serial, and (iii) using RSBB in parallel across 10 CPU cores. The solver is configured with a gap tolerance of 0.01 and a maximum runtime of 14,800 seconds. Each setup is repeated five times, and we report the mean and standard deviation of the runtime.

**Results.** Figure 4 summarizes the results. For all instances except `tic_vs_ran` and `tiger_vs_ant`, where none of the methods reach optimality within the time limit, parallel RSBB consistently achieves superior efficiency. For example, on the `firewire` benchmark, solving directly with Gurobi requires 11,528.12 s, whereas parallel RSBB completes the task in only 15.91 s. Similarly, on `csma_2_4`, Gurobi exceeds the 14,800 s limit, while parallel RSBB finishes in just 42.568 s. This dramatic reduction in computation time highlights the efficiency of the parallel RSBB approach.

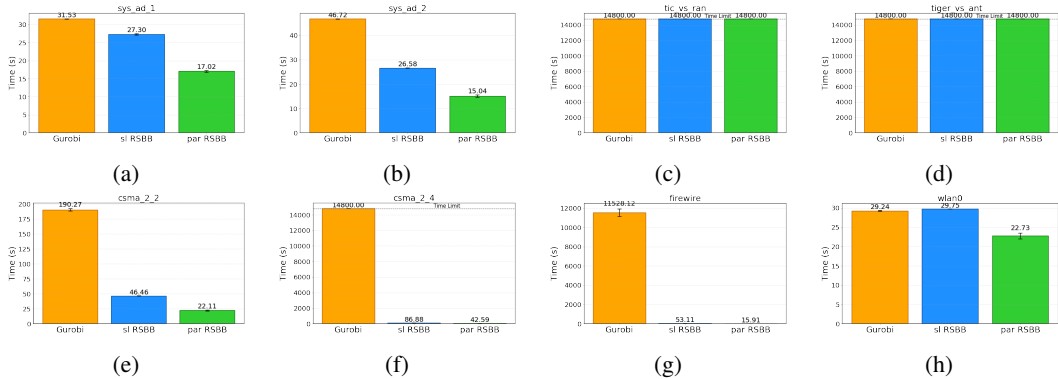

Figure 4: Running time comparisons across different MDPs using Gurobi, serial RSBB, and parallel RSBB. Each bar represents the mean runtime, with error bars showing standard deviation.

# D  Proofs

## D.1  Proof of Proposition 1

*Proof.* Given the current policy $\pi^{\text{old}}$, consider the optimization performed by SPOT at the update step:

$$\max_{\mu} \sum_i \Phi_i^{\pi^{\text{old}}} \sum_k \mu_{ik} \sum_{i'} P_{ii'k} \left( R_{ii'k} + \gamma V_{i'}^{\text{old}} \right).$$

subject to $\sum_k \mu_{ik} = 1$ for each state $i$ and $\mu_{ik} \geq 0$. Here $P_{ii'k}$ and $R_{ii'k}$ are the transition probability and reward for taking action $k$ in state $i$ and landing in state $i'$, and $V_{i'}^{\text{old}}$ is the value of state $i'$ under $\pi^{\text{old}}$. This objective is the expected total discounted return of the new policy evaluated under the state visitation weights of the old policy. To see this, define the $Q$-value of $\pi^{\text{old}}$ for state–action pair $(i, k)$

as

$$Q^{\pi^{\text{old}}}(i, k) = \sum_{i'} P_{ii'k} \left( R_{ii'k} + \gamma V_{i'}^{\text{old}} \right).$$

Using this definition, we can rewrite the objective in a simpler form:

$$\max_{\mu} \sum_{i} \Phi_i^{\pi^{\text{old}}} \sum_{k} \mu_{ik} Q^{\pi^{\text{old}}}(i, k).$$

This expression represents the expected $Q$-value of the new policy, weighting each state $i$ by its discounted occupancy $\Phi_i^{\pi^{\text{old}}}$ under $\pi^{\text{old}}$. Because $\Phi^{\pi^{\text{old}}}$ is fixed during this optimization, the objective separates by states. Maximizing it will assign, for each state $i$, all probability mass $\mu_{ik}$ to the action $k$ that has the highest $Q^{\pi^{\text{old}}}(i, k)$. In other words, the optimal $\mu$ is the greedy policy $\pi^{\text{new}}$ defined by $\pi^{\text{new}}(k|i) = 1$ for $k = \arg\max_a Q^{\pi^{\text{old}}}(i, a)$. This yields the maximum of the weighted $Q$-value objective.

To connect this update to the policy gradient, recall the policy gradient theorem [21]. For a differentiable policy $\pi_\theta$, the gradient of the expected return $J(\pi_\theta)$ can be written as

$$\nabla_\theta J(\pi_\theta) = \sum_{i} \Phi_i^{\pi_\theta} \sum_{k} \nabla_\theta \pi_\theta(k \mid i) Q^{\pi_\theta}(i, k).$$

In particular, at $\theta = \text{old}$ (i.e. at the current policy), the direction of steepest ascent in policy space is determined by the term $\sum_i \Phi_i^{\pi^{\text{old}}} \sum_k Q^{\pi^{\text{old}}}(i, k) \nabla_\theta \pi_\theta(k|i)$. Intuitively, this means that in each state $i$, increasing the probability of an action $k$ will increase $J$ at a rate proportional to $\Phi_i^{\pi^{\text{old}}} Q^{\pi^{\text{old}}}(i, k)$. The optimization we performed above uses this same signal: $\Phi_i^{\pi^{\text{old}}} Q^{\pi^{\text{old}}}(i, k)$ appears as the coefficient of $\mu_{ik}$ in the objective. Thus, choosing the action that maximizes $Q^{\pi^{\text{old}}}(i, k)$ for each state (i.e. setting $\mu_{ik} = 1$ at the maximizing $k$) is precisely what the policy gradient would prescribe — it favors actions with the largest positive impact on the objective. In fact, the above $\mu$-update can be seen as solving for the policy that maximizes the first-order expansion of $J(\pi)$ around $\pi^{\text{old}}$. By taking a greedy, full step in the direction of this gradient (rather than an infinitesimal step), SPOT produces the deterministic policy $\pi^{\text{new}}$ that locally maximizes the expected return improvement.

□

### D.2 Proof of Proposition 2

*Proof.* Assume that the decision tree policy class has enough capacity to represent the optimal policy. This means at each iteration, when SPOT optimizes the tree policy, it can effectively carry out a policy improvement step with respect to the current value function $V^{\text{old}}$. We show that this improvement step will monotonically increase the policy's performance and reach optimality.

More specifically, for a fixed $V^{\text{old}}$, the objective follows:

$$\max_{\mu} \sum_{i} \Phi_i^{\pi^{\text{old}}} \sum_{k} \mu_{ik} \sum_{i'} P_{ii'k} \left( R_{ii'k} + \gamma V_{i'}^{\text{old}} \right). \tag{9}$$

Because the maximization separates by state (note that $\mu_{ik}$ for each state $i$ appears only in the term for state $i$ ), we can optimize each state's decision independently. Let us define the Q-value of taking action $k$ in state $i$ under the old policy as

$$Q^{\pi^{\text{old}}}(i, k) = \sum_{i'} P_{ii'k} \left( R_{ii'k} + \gamma V_{i'}^{\text{old}} \right).$$

Maximizing the inner sum over actions for a given state $i$ will simply pick the action that maximizes this $Q$-value. In other words, for each state $i$,

$$\max_{\mu_i} \sum_{k} \mu_{ik} Q^{\pi^{\text{old}}}(i, k) = \max_{k} Q^{\pi^{\text{old}}}(i, k),$$

since the optimal choice is to put all the state's probability mass on the single action $k$ that yields the highest $Q^{\pi^{\text{old}}}(i, k)$ (this corresponds to choosing that action deterministically in state $i$ ). Therefore, the entire objective decouples over states and can be written as:

$$\sum_{i} \Phi_i^{\pi^{\text{old}}} \max_{k} Q^{\pi^{\text{old}}}(i, k).$$

The policy $\pi^{\text{new}}$ that achieves this maximum is precisely the greedy policy improvement over $\pi^{\text{old}}$: for each state $i$, $\pi^{\text{new}}(i) = \arg\max_k Q^{\pi^{\text{old}}}(i, k)$. By construction, this new policy $\pi^{\text{new}}$ is a decision tree (deterministic) policy, assuming the tree function approximator is flexible enough to represent that state-to-action mapping.

Crucially, because $\Phi_i^{\pi^{\text{old}}} > 0$ for every state $i$, no state is ignored in this optimization. The strictly positive weights ensure that improving the $Q$-value in any state will increase the overall objective. In other words, the weighted objective preserves the true ordering of policy performance: if one policy yields higher value at some state without lowering it elsewhere, the objective (with all-positive weights) will also be higher for that policy. This guarantees that the greedy step indeed aligns with improving the actual value function of the policy.

The described greedy update is exactly the policy improvement step from dynamic programming. By the policy improvement theorem [17], the new policy $\pi^{\text{new}}$ is guaranteed to be no worse than $\pi^{\text{old}}$ in terms of its value for all states, and in fact strictly better for at least one state if $\pi^{\text{old}}$ was not already optimal. In other words, $V^{\pi^{\text{new}}}(i) \geq V^{\pi^{\text{old}}}(i)$ for all states $i$, with a strict inequality for some state as long as $\pi^{\text{old}}$ is not optimal. Because our decision-tree function class can represent the policy exactly and SPOT guarantees solving the optimal policy in each step, it finds the optimal policy after convergence.

$\square$

### D.3 Proof of Theorem 1

The proof of Theorem 1 follows naturally from observing that the set of feasible solutions is finite. Specifically, since each decision node in the tree can only take a finite set of meaningful split values $b$, the number of distinct decision tree structures generated from these splits is also finite. Consequently, a finite enumeration of solutions is inherently guaranteed, making the convergence of the proposed RSBB algorithm straightforward.

Additionally, Problem (5) can be viewed as a specialized instance of a two-stage stochastic programming problem. Consequently, the convergence analysis for our RSBB algorithm can leverage established methodologies presented in earlier literature, particularly the frameworks described by [9] and [15]. Although our problem features a distinct cost function formulation tailored for reinforcement learning policy optimization, the fundamental structure of the branching process in SPOT closely mirrors that of the classification tree optimization problems studied in [15]. Specifically, the choice of branching variables in each iteration, such as the split indicator variables ($d$), state feature selection indicator variables ($a$), and threshold variables ($b$), remain identical to those in previous optimal decision tree frameworks. Therefore, despite variable $b$ being continuous rather than discrete, the convergence properties derived in earlier works can also be applied in the current problem. This continuity does not compromise convergence guarantees, as the underlying finite combinatorial structure induced by discrete structural variables ensures that the algorithm systematically explores a finite solution space and thus converges to the global optimum.

## E Branching Strategy for Algorithm 2

Algorithm 2's branching strategy prioritizes variables according to their structural significance within the optimal decision tree. The binary variables $d$, which directly encode whether individual decision nodes perform splits, naturally form the foundational structure of the tree. Therefore, branching decisions first focus on these split-indicator variables.

For the remaining variables $(a, b, c)$, we employ a heuristic procedure as follows: we first compute a branching threshold defined as $\tau = 1 - \frac{1}{2}\|b^u - b^l\|_\infty$. We then compare $\tau$ to a uniformly generated random number in the range $[0, 1]$. If this random number exceeds $\tau$, branching occurs on the discrete variable $a$, which specifies the splitting feature at the decision node. Should all components of $a$ already be fixed, we instead branch on the variable $c$, controlling the action assignment at leaf nodes. Alternatively, if the random number is less than or equal to $\tau$, branching is conducted on the continuous variable $b$, selecting the midpoint between its current bounds as the branching point.

Within each category of variables $(a, b, c)$, we systematically prioritize branching based on ascending node indices; for example, if two feature-selection variables $a_{j,t}$ and $a_{j,t+1}$ remain undetermined, branching will occur first on the variable associated with the lower-indexed node, i.e., $a_{j,t}$.

