# OpenReview forum: "SPOT: Scalable Policy Optimization with Trees for Markov Decision Processes"
_NeurIPS.cc/2025/Conference — NeurIPS 2025 poster_

### Official Review · Reviewer_LbzP · 2025-06-16

**Clarity:** 3
**Significance:** 4
**Originality:** 3
**Rating:** 5
**Confidence:** 3

**Summary:**

The authors study the problem of optimizing decision tree policies in Markov Decision Processes (MDPs).
In particular, the authors introduce SPOT, a novel framework for learning interpretable and high-performing decision tree policies in MDPs.
Unlike traditional black-box reinforcement learning models, SPOT directly optimizes decision tree structures using a mixed-integer linear programming formulation.
Theoretical and empirical results show that SPOT achieves fast runtimes and good performance across several benchmark MDPs, generating compact and interpretable policies even for large-scale problems.

**Questions:**

Page 1:
You argue that directly learning interpretable decision tree policies is preferable to post-hoc explanations of black-box models.
What are the key limitations you've observed with explanation methods like LIME or SHAP in RL settings?

Page 2:
Figure 1:
Please elaborate on the caption.
What is the main takeaway here?

Page 3:
Where is Section 1.1? :)

Page 4:
Can you please put formal statements of your results in the Introduction?

Page 5:
How does SPOT perform when the optimal policy cannot be well-approximated by a shallow decision tree?
Given that decision trees are inherently limited in expressiveness with small depth, it would be interesting to see how SPOT handles situations where a more complex policy is required.

Page 6:
The explanations are very good :)

I do not understand lines 200 -- 204.

Page 7:
How does the reduced-space branch-and-bound framework maintain solution quality while only branching on first-stage (tree structure) variables?

Page 8:
Can you please clarify the purpose of using the Reduced-Space Branch-and-Bound (RSBB) method in the SPOT algorithm? :)

How does the RSBB framework ensure scalability and convergence when optimizing decision-tree policies?

Page 9:
In your experiments, SPOT+WS significantly outperforms OMDT on some benchmarks.
Can you please elaborate on the characteristics of these environments that make SPOT particularly effective? :)

The Conclusion section is too small :)
Please elaborate on future work.

**Ethical Concerns:**

["NO or VERY MINOR ethics concerns only"]

**Final Justification:**

The authors have effectively addressed my comments and concerns.

**Limitations:**

None.

**Paper Formatting Concerns:**

None.

**Quality:**

4

**Strengths And Weaknesses:**

Strengths:
The theoretical guarantees of the main theorems.

Weaknesses:
See Questions below.

---

> ### Author Rebuttal · Authors · 2025-07-31
>
> We thank the reviewer for your thoughtful comments that helped us improve this paper. Below, we would like to provide point-to-point responses to your major comments mentioned and hope they are to your satisfaction.
>
> > Page 1: You argue that directly learning interpretable decision tree policies is preferable to post-hoc explanations of black-box models. What are the key limitations you've observed with explanation methods like LIME or SHAP in RL settings?
>
> We appreciate the reviewer's insightful question about the limitations of post-hoc explanation methods in RL settings. We agree that elucidating the precise shortcomings of these popular techniques in the RL setting strengthens the motivation for our work. Below we summarise the main limitations of methods like LIME and SHAP when applied to RL policies:
>
> - **Inconsistent across nearby states**: The stochastic nature of RL environments and policies can lead to unstable explanations across similar states, making it difficult to derive consistent insights about the policy's behavior.
> - **Ignores long‑term value**: Attribution is usually computed w.r.t. immediate Q‑value or probability of the chosen action, not the downstream return.
> - **Heavy computation cost**: Generating explanations for every state-action pair during deployment introduces significant computational costs.
> - **Non‑stationary policy distribution**: During training the data distribution shifts; explanations computed mid‑training quickly become obsolete.
>
> In contrast, our interpretable decision tree approach learns a global, consistent, and symbolic policy that can be directly inspected and validated by domain experts without incurring additional computational overhead.
>
> > Page 2: Figure 1: Please elaborate on the caption. What is the main takeaway here?
>
> Thank you for pointing this out. We will expand the caption to better convey the key message. Figure 1 demonstrates an interpretable policy learned by our decision tree approach, where each path from root to leaf represents a clear decision rule. The main takeaway is that our method produces policies that are directly interpretable—domain experts can trace any decision by following the tree structure, understanding exactly which features lead to specific actions, unlike black-box neural network policies that require post-hoc explanations.
>
> >Page 3: Where is Section 1.1? :)
>
> Thank you for catching this formatting issue. Section 1.1 begins on Line 38, page 2.
>
> >Page 5: How does SPOT perform when the optimal policy cannot be well-approximated by a shallow decision tree? Given that decision trees are inherently limited in expressiveness with small depth, it would be interesting to see how SPOT handles situations where a more complex policy is required.
>
> This is an excellent question that highlights a fundamental trade-off in our approach. Empirically, we found that even with limited expressiveness (e.g., depth $D=3$), SPOT can find competitive solutions for complex MDPs like `firewire` within the 60-minute computational budget. While shallow trees may not capture the full complexity of optimal policies in some domains, they often identify robust decision rules that perform well in practice. We acknowledge this limitation and plan to investigate in future work: (1) adaptive depth selection based on problem complexity, and (2) hybrid approaches that maintain interpretability while increasing expressiveness for domains requiring more complex policies.
>
> >I do not understand lines 200 -- 204.
>
> We apologize for the unclarity in our explanation. We will revise lines 200-204 to improve clarity. Specifically:
>
> - Constraints (4a) and (4b) both ensure that a decision tree leaf only represents one action (one policy one action) and one state can only corresponding to one unique policy.
> - When a branching node has a split ($d_t > 0$): Constraints (4c) and (4d) activate the standard branching logic—states go left if their feature value is below the threshold b_m, and right otherwise.
> - When a node has no split ($d_t = 0$): Constraints (4e) and (4f) ensure special handling. Since no branching occurs, all states reaching this node must proceed to the right branch. This is achieved by setting the threshold $b_t = 0$ (from 4f), which, combined with constraint (4d), forces all states rightward since any feature value $x_i ≥ 0$.
>
> More details could refer to the Appendix A of Hua et al. (2022) [1] and Section 8.2 of Bertsimas and Dunn (2019) [3]. We will also rewrite this section with clearer language and possibly add a small example to illustrate this branching logic more intuitively.
>
>
> >Page 7: How does the reduced-space branch-and-bound framework maintain solution quality while only branching on first-stage (tree structure) variables?
>
> Briefly speaking, at each node of our RSBB procedure, we construct a lower bound by relaxing the non-anticipativity constraints (i.e., allowing each sample to have its own tree model), while we compute a simple upper bound by fixing the first-stage decision variables to the current feasible solution. With these two bounding strategies, the convergence of RSBB is ensured by branching exclusively on the first-stage variables, while **branching on the second-stage variables is handled implicitly through the evaluation of the lower and upper bounds**. Detailed proofs can be found in Cao and Zavala (2019)[2] and Hua et al. (2022)[1].
>
> >Page 8: Can you please clarify the purpose of using the Reduced-Space Branch-and-Bound (RSBB) method in the SPOT algorithm? :)
>
> The RSBB method serves as a computational efficiency enhancement. It achieves the same optimal solution quality as standard branch-and-bound but significantly faster by exploiting the problem structure, **branching only on tree structure variables** while solving action assignments analytically. Moreover, both bounding strategies can be evaluated by independently solving scenario-specific subproblems, which enables efficient parallel computation. These efficiency improvements are crucial for the empirical success of SPOT on larger problems.
>
> >How does the RSBB framework ensure scalability and convergence when optimizing decision-tree policies?
>
> Our scalability are achieved through the reduce of branching variables and the trivial parallelism for solving the bounding problems in each BB node. As a systematic enumeration algorithm. The convergence is guaranteed by systematically exploring the solution space and dividing it into smaller subproblems, and using bounding functions to eliminate portions of the space that cannot contain the optimal solution.
>
> >Page 9: In your experiments, SPOT+WS significantly outperforms OMDT on some benchmarks. Can you please elaborate on the characteristics of these environments that make SPOT particularly effective? :)
>
> This is an insightful question. Due to the highly non-convex nature of the decision tree optimization problem and its complex loss landscape, we cannot yet provide definitive characteristics that predict when SPOT will excel. Our empirical results suggest SPOT performs particularly well in large scale MDP and where good warm-start solutions exist, but a thorough theoretical characterization remains an important direction for future work.
>
> >Page 4: Can you please put formal statements of your results in the Introduction?
> >The Conclusion section is too small :) Please elaborate on future work.
>
> Thank you for these constructive suggestions. We will revise our paper in the new version.
>
> **References**
>
> [1] Hua et al., "A Scalable Deterministic Global Optimization Algorithm for Training Optimal Decision Tree," *Advances in Neural Information Processing Systems (NeurIPS)*, 2022.
>
> [2] Cao, Y., & Zavala, V. M., "A scalable global optimization algorithm for stochastic nonlinear programs," *Journal of Global Optimization*, 2019.
>
> [3] Dimitris Bertsimas and Jack Dunn, "Machine learning under a modern optimization lens," *Dynamic Ideas LLC*, 2019.

---

### Official Review · Reviewer_CDVk · 2025-06-18

**Clarity:** 3
**Significance:** 3
**Originality:** 3
**Rating:** 5
**Confidence:** 4

**Summary:**

The paper presents a novel method for computing policies for Markov decision processes (MDPs) represented as decision trees.
In particular, the scalability issues arising in previous approaches are addressed through a new formulation of the underlying optimization problem. These MILPs can be decomposed, allowing for parallelized optimization.
An experimental evaluation compares the new approaches with a baseline.

**Questions:**

Q1. Why does SPOT outperform SPOT+WS on some benchmarks (sys_ad, tic_vs_ran, ...)? Is the initial solution used as a warm start worse than the standard initialization of SPOT in these cases?

**Ethical Concerns:**

["NO or VERY MINOR ethics concerns only"]

**Final Justification:**

I believe this paper is a strong contribution, and the authors have addressed all my concerns in the rebuttal. I continue to recommend acceptance.

**Limitations:**

Yes.

**Quality:**

3

**Strengths And Weaknesses:**

## Strengths

+ The paper is overall well-written.
+ The method addresses important limitations of previous work.
+ The key theoretical properties one would expect (optimality, convergence) are proven.
+ The experimental evaluation shows improvements over the state-of-the-art baseline on all benchmarks.


## Weaknesses

- There are some (minor) issues with parts of the technical presentation, in particular around decision trees and the MILP encoding.

In more detail:

There is a lack of preliminaries on decision trees. There is no clearly marked definition, and while the key ingredients are in the paper (l183-190),
I think the paper would become clearer and more self-contained if decision trees were briefly introduced, similarly to how MDPs are introduced.
In a similar vein, the decision trees rely on some set of features F that currently appear out of nowhere. It would make sense to connect the set of features to the MDPs you consider (i.e., via a factored state-space or a labeling function).

While the MILP formulation is rather straightforward, its presentation could be improved. In particular, constraints 4c and 4d are the two most complicated constraints that require a more detailed explanation beyond what's currently there. More explanation on what "appropriate branching behavior" and "directing states correctly" mean is necessary here.

I would also encourage making more explicit what symbols are variables (and their allowed domains) and what are constants.

## Typos / other comments

- Mention that additional experimental results may be found in the appendix.
- I find it a bit weird that Frozen Lake is used as an example in the introduction, but is not present in the experiments.
- l280: global -> globally
- Eq.9: undtm -> use a symbol like \bot
- l298: "We evaluate our algorithm by comparing it to OMDT [23] on large-scale MDPs." I suppose these MDPs are large-scale in the context of decision-tree policies, but in general, MDPs with |S| < 10000 should not be considered large-scale. Make this remark more accurate.
- l304: SPOT1 -> I think you meant something like "SPOT (Alg. 1)"
- The use of a warm start shows up out of nowhere in the experiments. It would be better to (briefly) discuss this option earlier in the paper.

---

> ### Author Rebuttal · Authors · 2025-07-31
>
> Thank you for your thoughtful and constructive review of our paper. We appreciate your positive assessment and the opportunity to clarify and strengthen our contributions.
>
> > There is a lack of preliminaries on decision trees. There is no clearly marked definition, and while the key ingredients are in the paper (l183-190), I think the paper would become clearer and more self-contained if decision trees were briefly introduced, similarly to how MDPs are introduced. In a similar vein, the decision trees rely on some set of features F that currently appear out of nowhere. It would make sense to connect the set of features to the MDPs you consider (i.e., via a factored state-space or a labeling function).
>
> We sincerely thank the reviewer for this valuable suggestion to enhance the paper’s clarity and self-containment. We will include detailed and formal preliminaries on optimal decision trees in Section 2 if the paper is accepted. The definition of our tree model primarily follows the formulations used in the optimal decision tree literature, particularly Hua et al. (2022) [1] and Bertsimas and Dunn (2019) [2]. In the following, we provide a brief overview.
>
> **Decision Trees for Policy Representation**
>
> A decision tree is a hierarchical structure that maps state features to actions through a sequence of binary tests. Formally, a decision tree of depth $D$ consists of:
>
> - **Decision nodes** $T_D = \{1, \ldots, \lfloor T/2 \rfloor\}$ where $T = 2^{D+1} - 1$
> - **Leaf nodes** $T_L = \{\lfloor T/2 \rfloor + 1, \ldots, T\}$
> - **Splitting rules**: Each decision node $t$ performs a test of the form $x_j \leq \theta_t$ for feature $j$ and threshold $\theta_t$
> - **Action assignments**: Each leaf node $\ell$ is assigned an action $a_\ell \in A$
>
> The decision tree policy $\pi(i)$ for state $i$ is determined by traversing from the root to a leaf based on the feature vector $x_i$, then returning the action assigned to that leaf.
>
> This formalization connects the abstract MDP states to the concrete tree structure through the feature mapping $\phi$, making the approach more transparent.
>
> > While the MILP formulation is rather straightforward, its presentation could be improved. In particular, constraints 4c and 4d are the two most complicated constraints that require a more detailed explanation beyond what's currently there. More explanation on what "appropriate branching behavior" and "directing states correctly" mean is necessary here.
>
> Thank you for this important feedback regarding the clarity of constraints (4c) and (4d). We refer these two constraints to the work of Hua et al. (2022) [1] and Bertsimas and Dunn (2019) [2]. We will provide much more detailed explanation in the paper once accepted.
>
> **Constraint (4c) - Left Branch Conditions:**
> $a^T_m(x_i + ε_{im} - ε_{min}) + ε_{min} ≤ b_m + (1 + ε_{max})(1 - z_{it})$
>
> - **When $z_{it} = 1$** (state i assigned to leaf t): Enforces $a^T_m x_i ≤ b_m$ for all ancestor nodes m where the path goes left
> - **When $z_{it} = 0$**: Big-M term deactivates the constraint
>
> **Constraint (4d) - Right Branch Conditions:**
>  $a^T_m x_i ≥ b_m - (1 - z_{it})$
>
> - **When $z_{it} = 1$**: Enforces $a^T_m x_i ≥ b_m$ for all ancestor nodes m where the path goes right
> - **When $z_{it} = 0$**: Constraint is relaxed
>
> These constraints ensure **path consistency**: if state $i$ is assigned to leaf $t$, its features must satisfy every splitting condition along the root-to-leaf path. This prevents impossible assignments where a state reaches a leaf through a path inconsistent with its feature values. A detailed reference of the seting of $\epsilon$ can check Appendix A of Hua et al. (2022)[1] and Chapter 8.2 of Bertsimas and Dunn (2019) [2] .
>
> > I would also encourage making more explicit what symbols are variables (and their allowed domains) and what are constants.
>
> Thank you for this important feedback. We acknowledge that our notation could be much clearer in distinguishing between optimization variables and problem constants. The current presentation intermixes decision variables (e.g., $a_{jt} \in {0,1}$, $b_t \in [0,1]$, $\mu_{ik} \in [0,1]$) with fixed MDP parameters (e.g., $P_{i,i',k}$, $R_{i,i',k}$, $\gamma$) and algorithm constants (e.g., $\epsilon$, $\Phi_i^{\pi^{\text{old}}}$).
> | Category | Description | Example Variables |
> |----------|-------------|-------------------|
> | **Decision Variables**| Optimization variables to be solved | $a_{jt} \in \{0,1\}$, $b_t \in [0,1]$, $\mu_{ik} \in [0,1]$ |
> | **MDP Parameters**| Fixed transition probabilities and rewards | $P_{i,i',k}$, $R_{i,i',k}$, $\gamma$ |
> | **Algorithm Constants**| Fixed hyperparameters and computed values | $\epsilon$, $\Phi_i^{\pi^{\text{old}}}$ |
> | **State/Action Indices**| Index sets for states and actions | $i, i' \in \mathcal{S}$, $k \in \mathcal{A}$ |
> | **Time Indices**| Temporal or iteration indices | $t$|
>
>
> > I find it a bit weird that Frozen Lake is used as an example in the introduction, but is not present in the experiments.
>
> Thank you for this observation. Frozen Lake was chosen as an illustrative example in the introduction due to its simple structure that clearly demonstrates how decision tree policies work. However, in the experiments, we focus on larger, more challenging MDPs where SPOT's scalability advantages over OMDT are most apparent.
>
> > Mention that additional experimental results may be found in the appendix.
> > l280: global -> globally
> > Eq.9: undtm -> use a symbol like \bot
> > l298: "We evaluate our algorithm by comparing it to OMDT [23] on large-scale MDPs." I suppose these MDPs are large-scale in the context of decision-tree policies, but in general, MDPs with |S| < 10000 should not be considered large-scale. Make this remark more accurate.
> l304: SPOT1 -> I think you meant something like "SPOT (Alg. 1)"
> > The use of a warm start shows up out of nowhere in the experiments. It would be better to (briefly) discuss this option earlier in the paper.
>
> Thank you for these helpful presentation suggestions, we will address them in the new version.
>
> > Q1. Why does SPOT outperform SPOT+WS on some benchmarks (sys_ad, tic_vs_ran, ...)? Is the initial solution used as a warm start worse than the standard initialization of SPOT in these cases?
>
> Thank you for this excellent question. Warm start does not guarantee better performance compared to standard initialization - it simply provides a potentially good starting point. The effectiveness depends on the problem's characteristics and the search heuristics employed.
>
> Our algorithm performs gradient-like updates at each iteration, but the optimization landscape is highly non-convex with multiple local optima. It's possible that the standard random initialization lands in a region with better local geometry than the OMDT warm start solution, leading to superior final solutions within the same time budget. Additionally, the warm start solution from OMDT (generated under a 5-minute constraint) may not always represent the best starting point for SPOT's iterative refinement process.
>
> This phenomenon is not uncommon in non-convex optimization where different initialization strategies can lead to different local optima, and the "better" starting point is problem-dependent.
>
> **References**
> [1] Hua et al., "A Scalable Deterministic Global Optimization Algorithm for Training Optimal Decision Tree," *Advances in Neural Information Processing Systems (NeurIPS)*, 2022.
>
> [2] Dimitris Bertsimas and Jack Dunn, "Machine learning under a modern optimization lens," *Dynamic Ideas LLC*, 2019.

---

> > ### Comment · Reviewer_CDVk · 2025-08-04
> >
> > Thank you for your clarifications. I believe adding these to a final version of the paper would make an already strong contribution even better, and I hope (and assume) the authors will do so based on their reply.

---

> > > ### Author Response · Authors · 2025-08-09
> > >
> > > We appreciate your thoughtful feedback and support in improving the quality of our paper. We will incorporate these suggestions in our revision.

---

### Official Review · Reviewer_3Gn1 · 2025-06-29

**Clarity:** 2
**Significance:** 1
**Originality:** 3
**Rating:** 3
**Confidence:** 3

**Summary:**

This paper proposes **Scalable Policy Optimization with Trees** (SPOT), a method for learning optimal decision tree policies in Markov Decision Processes (MDPs). SPOT formulates policy optimization as a sequence of mixed-integer linear programs (MILPs), decoupling the tree structure from the MDP dynamics via a reduced-space branch-and-bound (RSBB) framework. Empirical results on several discrete MDP benchmarks show that SPOT outperforms a prior MILP-based method (OMDT).

**Questions:**

NA

**Ethical Concerns:**

["NO or VERY MINOR ethics concerns only"]

**Final Justification:**

This paper may have some interesting ideas, but I don't believe the empirical results and claims align. The references and content added in the rebuttal should be reviewed again before accepting the paper. Therefore, I maintain my original score.

**Limitations:**

The author only mentioned "*Its performance still relies on the efficiency of the underlying optimization solver, which may restrict scalability for extremely large or complex MDPs*" in Section 6.

**Quality:**

2

**Strengths And Weaknesses:**

## Strengths

- The algorithm design seems reasonable.
- The paper provides convergence guarantees under certain conditions.
- SPOT demonstrates policy quality on several benchmark MDPs compared to OMDT.

## Weaknesses

- **Narrow relevance to modern RL**: The paper is positioned as an interpretable RL contribution, yet it operates in a highly constrained setting (small tabular MDPs, decision trees). It does not engage with current trends in deep RL, offline RL, or neuro-symbolic methods, limiting its broader impact for the NeurIPS community.
- **Sparse and outdated related work**: This area seems very niche, and the literature review is narrow, with very few references from the past 2--3 years and minimal engagement with recent interpretable RL or explainable AI research. There are only 23 references in this paper. Only one from 2024, one from 2023, and two from 2022. Only one from NeurIPS. The paper could benefit from positioning itself more clearly within the broader ML community.
- "*Decision tree policies have attracted significant attention as a suitable interpretable model class*": This is very questionable and debatable. First, no references. "Significant attention" must be supported by sufficient references. Second, I don't think decision trees are inherently interpretable. "Simulatable by a person" is not a requirement for interpretability. One can "run" a neural network by hand, given enough time. Nevertheless, the experiments didn't show how we can interpret the policy anyway. No interpretability analysis, user study, or policy explanation is presented in experiments to support this core motivation.
- The theoretical convergence guarantee assumes that the class of decision tree policies can represent the optimal policy. This is a very strong and often false assumption. This contradicts other parts of the paper where the authors acknowledge that small trees are fundamentally limited in expressivity.
- "*This significantly improves runtime and scalability compared to previous methods*": I can't observe this from the experimental results. Since "*all experiments were performed with a uniform 60-minute computational budget.*" If the author wants to show speedup, they should show how long it takes to achieve the same performance. "OMDT (5m)" is unnecessary and unfair.

---

> ### Author Rebuttal · Authors · 2025-07-31
>
> > Narrow relevance to modern RL: The paper is positioned as an interpretable RL contribution, yet it operates in a highly constrained setting (small tabular MDPs, decision trees). It does not engage with current trends in deep RL, offline RL, or neuro-symbolic methods, limiting its broader impact for the NeurIPS community.
>
> We thank the reviewer for the valuable feedback. We respectfully argue that our focus on tabular MDPs and decision trees is not a limitation, but a choice to tackle a fundamental problem: finding provably optimal policies within an interpretable model class [2]. This pursuit is critical for building trustworthy AI systems suitable for high-stakes domains where safety and verifiability are paramount. Far from being disconnected from modern RL [1,6], our work engages directly with its key challenges. First, we address the well-documented "imitation gap," a known issue where distilling complex deep RL policies is often suboptimal compared to direct optimization of the interpretable model itself. Second, tree-based policies are highly relevant to modern offline RL, where recent work has shown that they offer remarkable training speed and stability by reframing the problem as regression [3]. Third, our method provides a foundational, verifiable building block for neuro-symbolic systems, which require high-quality symbolic components to ensure reliability [4]. Finally, our work contributes to what has been identified as a "grand challenge" in machine learning: the scalable optimization of sparse logical models such as decision trees [5].
>
>
> > Sparse and outdated related work: This area seems very niche ... The paper could benefit from positioning itself more clearly within the broader ML community.
>
> We thank the reviewer for the feedback. We respectfully disagree with the characterization of the related work as sparse or outdated and wish to clarify our work's position within the contemporary research landscape. The pursuit of inherently interpretable policies, such as decision trees, represents a major modern thrust in Explainable AI [6], particularly for safety-critical sequential decision-making. Our work engages with a vibrant and diverse field of recent tree-based RL methods, including gradient-boosted ensembles (e.g., GBRL [7]), concept-based learning (e.g., LICORICE [8]), and direct optimal policy search [2], where our work is situated. Our direct predecessor, OMDT [2], established the use of monolithic MILP formulations for this task. The primary contribution of our paper, SPOT, is to overcome the critical scalability bottleneck of this prior work. Our iterative decomposition and reduced-space branch-and-bound approach is not an ad-hoc heuristic but rather a novel application of a state-of-the-art optimization pattern that mirrors recent advances in the broader Machine Learning for Mixed-integer Optimization community (e.g., Apollo-MILP [9]). This methodology allows us to scalably produce globally optimal, size-constrained policies. This is a crucial capability, as single optimal trees are amenable to efficient formal verification [10], whereas verifying tree ensembles is known to be NP-complete [11], making our approach highly relevant for developing verifiably robust and trustworthy AI systems.
>
> We acknowledge the reviewer’s point that the paper would benefit from engaging more extensively with recent literature, especially from the last two years and from top venues such as NeurIPS. We will revise the manuscript to better position our work within the landscape.
>
> > "Decision tree policies have attracted significant attention as a suitable interpretable model class" ... No interpretability analysis, user study, or policy explanation is presented in experiments to support this core motivation.
>
> We respectfully disagree with the reviewer’s claim that decision trees lack “significant attention” and inherent interpretability, as this perspective appears to overlook a substantial body of literature. Decision tree policies have received considerable attention in reinforcement learning [2，12], with breakthrough work VIPER (NeurIPS 2018) [13] establishing the field, followed by developments including OMDT (IJCAI 2023) [2]. Decision trees are inherently interpretable due to their simulatability and decomposability properties [14], as established by foundational works including Breiman et al.'s CART [15], Quinlan's ID3 algorithm [16], and modern interpretability frameworks by Lipton [14]. Unlike neural networks, decision trees enable complete mental simulation of the decision process and provide "extraction of human-readable decision rules with full verification of input-output mapping" [17]. Empirical user studies consistently demonstrate superior interpretability of decision trees over neural networks [18,19], with quantitative evidence showing significantly higher interpretability ratings (p < 0.001) [12]. Rudin et al. identified "optimizing sparse logical models such as decision trees" as the #1 grand challenge in interpretable ML [6], distinguishing inherently interpretable models from post-hoc explanations. While we acknowledge the limitation regarding interpretability analysis in our experiments, the established theoretical foundations and extensive RL literature clearly support decision trees as a suitable interpretable model class for policy learning.
>
> >The theoretical convergence guarantee assumes that the class of decision tree policies can represent the optimal policy ... where the authors acknowledge that small trees are fundamentally limited in expressivity.
>
> We thank the reviewer for this important observation. The reviewer is correct that Proposition 2's assumption requiring decision tree policies to represent the optimal policy is strong and often unrealistic. To clarify: Proposition 2 establishes that SPOT converges to the globally optimal policy within the constrained decision tree policy space, not the unconstrained MDP optimum. In practice, we expect this assumption to be violated for most problems with small trees, but this doesn't diminish SPOT's value since it still finds the best possible tree policy given size constraints while avoiding poor local optima within the tree space. Moreover, the resulting policies are more robust and interpretable.
>
> We acknowledge that this limitation is inherent to all optimal decision tree methods (including OMDT). SPOT’s advantage lies in its computational efficiency for identifying the optimal tree, rather than in overcoming the expressivity constraint.
>
>
> >"This significantly improves runtime and scalability compared to previous methods" ... "OMDT (5m)" is unnecessary and unfair.
>
> We thank the reviewer for this important point. The reviewer is correct that our main experiments with uniform 60-minute budgets do not directly demonstrate speedup. We address this with dedicated runtime experiments **in Appendix B.3**, which compare solving identical optimization problems using (i) Gurobi directly, (ii) serial RSBB, and (iii) parallel RSBB. These results show substantial speedups for large problems—for example, parallel RSBB solves the firewire benchmark in 15.91 seconds versus 11,528 seconds for Gurobi, and completes csma_2_4 in 42.57 seconds while Gurobi exceeds the 14,800-second limit. We agree that the main text should better emphasize these runtime comparison results to support our speedup claims.
>
> **References**
>
> [1] Glanois et al., "A survey on interpretable reinforcement learning," *ACM Computing Surveys*, 2024.
>
> [2] Vos, D., & Verwer, S., "Optimal Decision Tree Policies for Markov Decision Processes," *Proceedings of IJCAI*, 2023.
>
> [3] Koirala, R., et al., "Decision Tree Policies for Offline Reinforcement Learning," *Conference on Robot Learning (CoRL)*, 2024.
>
> [4] Delfosse, Q., et al., "Interpretable and Explainable Logical Policies via Neurally Guided Symbolic Abstraction," *Advances in Neural Information Processing Systems (NeurIPS)*, 2023.
>
> [5] Rudin, C., et al., "Interpretable Machine Learning: Fundamental Principles and 10 Grand Challenges," *Statistics Surveys*, 2022.
>
> [6] Rudin, C., "Stop explaining black box machine learning models for high stakes decisions and use interpretable models instead," *Nature Machine Intelligence*, 2019.
>
> [7] Liu, C., et al., "Gradient Boosting Reinforcement Learning," *Proceedings of ICML*, 2025.
>
> [8] Ye, Z., et al., "LICORICE: Label-Efficient Concept-Based Interpretable Reinforcement Learning," *Proceedings of ICLR*, 2025.
>
> [9] Liu, H., et al., "Apollo-MILP: An Alternating Prediction-Correction Neural Solving Framework for Mixed-Integer Linear Programming," *Proceedings of ICLR*, 2025.
>
> [10] Kantaros, Y., et al., "Formal Verification of Decision-Tree-Controlled Continuous-Time Systems," *Advances in Neural Information Processing Systems (NeurIPS)*, 2023.
>
> [11] Kook, J., et al., "On the Computational Hardness of Interpreting Ensemble Models," *Proceedings of ICML*, 2025.
>
> [12] Silva, A., et al., "Optimization Methods for Interpretable Differentiable Decision Trees Applied to Reinforcement Learning," *Proceedings of AISTATS*, 2020.
>
> [13] Bastani, O., et al., "Verifiable Reinforcement Learning via Policy Extraction," *Advances in Neural Information Processing Systems (NeurIPS)*, 2018.
>
> [14] Lipton, Z. C., "The Mythos of Model Interpretability," *Communications of the ACM*, 2018.
>
> [15] Breiman, L., et al., "Classification and Regression Trees," *Routledge*, 1984.
>
> [16] Quinlan, J. R., "Induction of Decision Trees," *Machine Learning*, 1986.
>
> [17] Loh, W. Y., "Decision trees: From efficient prediction to responsible AI," *PMC*, 2022.
>
> [18] Huysmans, J., et al., "An empirical evaluation of the comprehensibility of decision table, tree and rule based predictive models," *Decision Support Systems*, 2011.
>
> [19] Bell, A., et al., "It's Just Not That Simple: An Empirical Study of the Accuracy-Explainability Trade-off in Machine Learning for Public Policy," *Proceedings of FAccT*, 2022.

---

> > ### Comment · Reviewer_3Gn1 · 2025-08-05
> >
> > Thank you for your detailed response.
> >
> > > finding provably optimal policies within an interpretable model class
> > > the scalable optimization of sparse logical models such as decision trees
> >
> > Again, I don't think trees are automatically more interpretable.
> >
> > In [1] Glanois et al. the author provided, they state that
> >
> > > A model is simulatable if its inner working can be simulated by a human. Examples of simulatable models are small linear models or decision trees. The concept of simplicity, and quantitative aspects, consequently underlie any definition of simulatability. In that sense, a hypothesis class is not inherently interpretable with respect to simulatability. Indeed, a decision tree may not be simulatable if its depth is huge, whereas a neural network may be simulatable if it has only a few
> > hidden nodes. A model is decomposable if each of its parts (input, parameter, and calculation) can be understood intuitively. Since a decomposable model assumes its inputs to be intelligible, any simple model based on complex highly-engineered features is not decomposable. Examples of decomposable models are linear models or decision trees using interpretable features.
> >
> > The point is simulatability $\neq$ interpretability. If the author claims that the proposed method is more interpretable, even in large-scale settings, they should provide empirical evidence or at least a case study to support this claim. **A table of normalized returns is not sufficient**.

---

> > > ### Author Response · Authors · 2025-08-09
> > >
> > > Thank you for clarifying this point. We agree that simulatability and interpretability are distinct, and that a hypothesis class is not inherently interpretable. We also believe that decision trees can support interpretability through qualities such as transparency and simulatability. When the tree depth is moderate and the features are not overly complex, this structure can produce a readily interpretable policy.
> > >
> > > We will also soften the language throughout the paper (removing any implication that decision trees are automatically interpretable) and instead position our contribution as delivering transparent, decomposable optimal policies, rather than interpretability in the broad sense.
> > >
> > > We also present the learned decision tree policy for the `tiger_vs_antelope` environment with depth $D=3$.
> > >
> > > ## Case Study
> > >
> > > The `tiger_vs_antelope` environment is a 5×5 grid world in which a tiger pursues an antelope. The state is the tuple antelope_x, antelope_y, tiger_x, tiger_y, with features normalized to [0,1]. The tiger can move up, right, down, left, or wait. The antelope moves randomly among valid cells, never leaving the grid or stepping onto the tiger’s current cell. The agent receives reward 1 when the tiger catches the antelope (same cell) and 0 otherwise, and the episode ends upon capture, and rewards are discounted with factor $\gamma\in(0,1)$. The task is challenging because the agent must anticipate stochastic antelope motion and use positioning to restrict escape routes
> > >
> > >
> > >
> > > ### Learned Tree Policy
> > > ```
> > > antelope_x ≤ 0.2?
> > > ├── Yes:
> > > │   tiger_x ≤ 0.6?
> > > │   ├── Yes:
> > > │   │   antelope_y ≤ 0.6  → wait
> > > │   │   antelope_y > 0.6  → up
> > > │   └── No: → left
> > > └── No:
> > >     antelope_y ≤ 0.8?
> > >     ├── Yes:
> > >     │   tiger_y ≤ 0.4  → up
> > >     │   tiger_y > 0.4  → down
> > >     └── No:
> > >         antelope_x ≤ 0.8  → up
> > >         antelope_x > 0.8  → right
> > >
> > > ```
> > >
> > >
> > > We will demonstrate the interpretability of the above learned policy through the following aspects:
> > >
> > > **1. Domain-Meaningful Behavior (which shows the strategical coherence)**
> > >
> > > From the learned decision tree policy, we summarize the following strategies:
> > >
> > > **Cornering Strategy**: When antelope is trapped near the left wall (x ≤ 0.2), the tiger adapts:
> > >
> > > * If tiger is well-positioned (x ≤ 0.6), it waits patiently or chases vertically
> > > * If tiger is far away (x > 0.6), it prioritizes closing the horizontal gap
> > >
> > > **Pursuit Strategy:** In open areas, the tiger uses intelligent vertical positioning:
> > >
> > > * Moves upward when positioned below the prey (tiger_y ≤ 0.4)
> > > * Moves downward when positioned above the prey (tiger_y > 0.4)
> > >
> > > **Escape Prevention**: Near top boundary (antelope_y > 0.8), the policy focuses on blocking escape routes rather than direct pursuit
> > >
> > >
> > > **2. Addresses Complexity Concern (This pattern collerate to the concept of Simulatability in Glanois et al.)**
> > >
> > > * Only 8 decision rules cover the entire 625-state space (5⁴ states)
> > > * Each path through the tree corresponds to a clear strategic situation
> > > * The shallow depth (D=3) ensures human comprehension
> > >
> > >
> > > **3. Uses Interpretable Features (This pattern collerate to the concept of Decomposability in Glanois et al.)**
> > >
> > > * Spatial boundaries: Recognizes walls (x ≤ 0.2 for left edge, y > 0.8 for top)
> > > * Relative positioning: Implicitly reasons about tiger-antelope spatial relationships
> > > * No complex engineered features: Uses raw normalized coordinates that directly map to grid positions.

---

> > ### Comment · Reviewer_3Gn1 · 2025-08-05
> >
> > I appreciate the added references.
> > This paper would benefit from better contextualization.
> > Every sentence should be supported by references or your own evidence provided in the paper.

---

> > > ### Comment · Reviewer_iZpk · 2025-08-05
> > > **Interpretability**
> > >
> > > My personal opinion is that there is great value in the decision maker’s ability to interpret and understand the optimal policy.

---

> > > > ### Author Response · Authors · 2025-08-09
> > > >
> > > > We appreciate your perspective on the importance of interpretability to decision makers.

---

> > > ### Author Response · Authors · 2025-08-09
> > >
> > > Thank you for this valuable feedback. We agree that proper contextualization will strengthen the paper’s arguments, and we will ensure that all claims are supported by appropriate references or empirical evidence in the final version.

---

### Official Review · Reviewer_iZpk · 2025-06-30

**Clarity:** 2
**Significance:** 3
**Originality:** 3
**Rating:** 5
**Confidence:** 3

**Summary:**

This paper develops a decision tree representation for the optimal policy to a problem represented as a Markov Decision Process (MDP), in order to be as transparent as possible for decision makers applying the policy.
This is a combinatorially complex problem, so the solution method applies a combination of techniques to arrive at an efficient solution, including mixed-integer linear programming (MILP) and a reduced-space branch-and-bound (RSBB) framework.
The decision tree optimization and the MDP dynamics are artfully decoupled so that significant subtasks can be performed in parallel.
The authors claim to achieve a more interpretable and scalable solution to deliver optimal policies an order of magnitude faster than existing approaches.

**Questions:**

How many processors would be needed to obtain the reported speed ups for csma and firewire?

Why do you report the “best gain” of the regular algorithm with or without warm start on the alternative algorithm as opposed to the gain from each?  You are considering whichever method works better, and even relying on the alternative algorithm for a warm start in order to show any improvement over that alternative algorithm, such as for wlan0 and csma_2_2 in both Table 1 and Table 2.

Are there any other limitations to your results besides the concerns about the efficiency of the underlying optimization solver?

Your algorithm compares very well as described when parallel processors can be employed for some test problems, but not others, and sometimes with a warm start, and sometimes without a warm start.  Why do you overstate its relative performance in general?

**Ethical Concerns:**

["NO or VERY MINOR ethics concerns only"]

**Final Justification:**

I found the insights from the reviewers and the responses from the authors to be quite helpful.

I believe that the paper’s emphasis on decision trees for interpretability of the optimal policy makes sense, but that the paper can still be improved based on many of the reviewers’ suggestions.

**Limitations:**

These are challenging problems and, although the innovations introduced appear to make a difference for some problems, the optimization solver is not the only limitation.

This seems to be promising work in progress, and I am concerned that the benefits in general are overstated.

**Quality:**

3

**Strengths And Weaknesses:**

The algorithm is complex with multiple reframings and transformations, and it is rather hard to follow.  Many of the details of the innovations to previous work, including the RSBB framework, are only fully described in the appendix.

The innovations to allow parallel processing appear to be original, but their significance seems overstated.  While the new algorithm outperforms the alternatives on multiple test datasets, on others the improvement cannot be characterized as “an order of magnitude faster than existing approaches.”  It does perform better when compared with tree depth 4 than with tree depth 3, and speed up with parallel processing is sometimes more significant than other times, but I could not find how many processors it would take to achieve that speed up.

---

> ### Author Rebuttal · Authors · 2025-07-31
>
> Thank you for your thoughtful and constructive review of our SPOT paper. We appreciate your detailed feedback and address each of your major concerns below.
>
> > How many processors would be needed to obtain the reported speed ups for csma and firewire?
>
> All experiments reported in this paper, including those for csma and firewire, were conducted using 10 CPU cores, as detailed in Appendix B.3. This hardware configuration was maintained consistently across all benchmark problems to ensure fair and reliable comparisons.
>
>
> > Why do you report the “best gain” of the regular algorithm with or without warm start on the alternative algorithm as opposed to the gain from each? You are considering whichever method works better, and even relying on the alternative algorithm for a warm start in order to show any improvement over that alternative algorithm, such as for wlan0 and csma_2_2 in both Table 1 and Table 2.
>
> You raise a valid point about clarity in our reporting. While we show individual performance for SPOT and SPOT+WS, we will revise the tables to report separate performance gains for each method instead of only the "best gain."
> Regarding the warm-start setup: SPOT+WS uses the 5-minute solution from OMDT as initialization but is compared against OMDT’s full 60-minute solution. We think this constitutes a fair comparison, as it leverages a lightweight version of a standard optimization technique to achieve improved performance within the same overall computational budget.
>
> > Are there any other limitations to your results besides the concerns about the efficiency of the underlying optimization solver?
>
> > These are challenging problems and, although the innovations introduced appear to make a difference for some problems, the optimization solver is not the only limitation.
>
> We agree that our method also has limitations. First, it cannot guarantee global optimality when the decision tree is not sufficiently deep to represent the true optimal policy. While our algorithm is applicable to deeper trees, solver limitations may prevent us from obtaining globally optimal solutions in such cases. Second, for small problems, the baseline OMDT can often find high-quality solutions quickly, leaving limited room for further improvement regardless of our algorithmic advances. That said, scalability remains a key challenge in interpretable reinforcement learning. As shown in Tables 1 and 2, OMDT struggles with problems involving more than 1,000 states, whereas our method maintains strong performance at this scale.
>
> > Your algorithm compares very well as described when parallel processors can be employed for some test problems, but not others, and sometimes with a warm start, and sometimes without a warm start. Why do you overstate its relative performance in general?
>
> We appreciate the reviewer's careful examination of our results. We would like to clarify several points:
>
> - **Parallel processing is used consistently for SPOT:** To ensure a fair comparison, all experiments were run on identical hardware, giving every method (OMDT, SPOT, and SPOT+WS) access to the same multi-core processors. The parallelization in our SPOT algorithm is an inherent feature of its branch-and-bound design, not a selective advantage. This parallel implementation yield speedup compared to its serial counterpart, as detailed in Appendix B.3.
> - **Performance across different MDPs:** We acknowledge that performance gains vary across test problems. This variation is due to the diverse characteristics of the MDPs (ranging from 256 to 7958 states). We report all results transparently, including cases where improvements are marginal (e.g., wlan0 with 0% gain) to provide a complete picture.
> - **Warm start vs. Cold start:** We evaluate both variants to demonstrate our method's versatility. Empirically, we observe that for larger MDPs (like `firewire, csma_2_4, wlan0, wlan1`),  SPOT with warm start consistently achieves superior performance.
>
> We will revise our claims to more precisely state that "SPOT consistently improves or matches baseline performance across diverse MDPs, with particularly strong gains on complex instances," rather than implying universal superiority.
>
> > This seems to be promising work in progress, and I am concerned that the benefits in general are overstated.
>
> We respectfully disagree with the characterization as "work in progress" and would like to address the concern about overstated benefits:
>
> * **Complete methodology:** Our work presents a fully developed algorithmic framework with (i) theoretical foundations for the policy iteration scheme, (ii) a complete branch-and-bound algorithm with proven correctness, and (iii) comprehensive empirical validation on established benchmarks.
> * **Realistic benefit reporting:** We report the full spectrum of results - from marginal improvements (0.2% on sys_ad_1) to substantial gains (1201.7% on firewire). This reporting demonstrates that we are not cherry-picking results. The "best gain" column shows the actual improvements achieved.
> * **Numerical robustness:** Our experiments evaluate multiple benchmark MDPs across varying problem sizes to ensure robust conclusions. The consistent improvements observed across diverse problem structures validate the general applicability of our approach.
>
> We acknowledge that our abstract and introduction may have used overly broad language. We will revise these sections to more accurately reflect that "SPOT provides solid improvements on challenging MDPs while maintaining competitive performance on simpler instances," which better represents our empirical findings.

---

### Note · Authors · 2025-08-13

In this work, we proposed SPOT, a novel and scalable optimization framework for learning interpretable decision tree policies in Markov Decision Processes (MDPs). Our key innovation lies in formulating policy optimization as a mixed-integer linear program combined with a reduced-space branch-and-bound (RSBB) approach, which decouples MDP dynamics from tree constraints and enables efficient parallel computation. This design significantly improves scalability and runtime compared to prior MILP-based methods, achieving near-optimal policies for MDPs with thousands of states while maintaining policy interpretability through compact tree structures.

Reviewers acknowledged several strengths of our submission: (1) the soundness and theoretical rigor including convergence and optimality guarantees, (2) the empirical demonstration of superior performance and scalability over the state-of-the-art OMDT baseline on diverse benchmark MDPs, and (3) the clear articulation and relevance of interpretability in high-stakes, safety-critical reinforcement learning domains.

During the rebuttal stage, we made substantial clarifications and enhancements: (1) we provided detailed explanations on the technical novelties of the RSBB framework and MILP constraints to improve clarity, (2) we addressed concerns about interpretability by illustrating the learned decision tree policies and relating them to established interpretability concepts such as simulatability and decomposability, including a case study, and (3) we transparently discussed limitations regarding tree depth expressiveness and computational trade-offs, alongside empirical evidence supporting runtime speedups and fair experimental comparisons. These efforts comprehensively addressed reviewers’ concerns, reinforcing the significance and robustness of our approach.

We believe that SPOT advances the foundational understanding and practical capabilities of interpretable reinforcement learning by enabling globally optimal, scalable, and verifiable policy synthesis through decision trees, thus contributing meaningfully to responsible AI in safety-critical applications.

---

### Decision · Program_Chairs · 2025-09-17

**Decision:**

Accept (poster)

**Comment:**

Three out of four reviewers (iZpk, CDVk, LbzP) were ultimately positive, recognizing the method's technical soundness, theoretical guarantees, and empirical improvements over baselines. Reviewer CDVk noted it as "a strong contribution" with proven optimality and convergence properties, while Reviewer LbzP praised the "excellent" theoretical guarantees and significance.

The main dissenting voice was Reviewer 3Gn1, who raised concerns about narrow relevance to modern RL, sparse literature review, and questionable interpretability claims. However, the authors provided comprehensive rebuttals addressing these concerns, including expanded literature positioning, a detailed interpretability case study, and clarification of theoretical assumptions.

While Reviewer 3Gn1 maintained their borderline reject position, the other reviewers found the work technically solid with clear contributions to interpretable RL. The method addresses an important problem—finding provably optimal, interpretable policies—which is valuable for safety-critical applications even if not directly applicable to deep RL settings.

The work provides novel algorithmic contributions through its reduced-space branch-and-bound approach, demonstrates scalability improvements over existing methods, and maintains theoretical rigor with convergence guarantees.